# PILAF: Optimal Human Preference Sampling for Reward Modeling

**Yunzhen Feng** [1]   **Ariel Kwiatkowski** [2] [*]   **Kunhao Zheng** [2] [*]   **Julia Kempe** [2] [1] [◇]   **Yaqi Duan** [1] [◇]

## Abstract

As large language models increasingly drive real-world applications, aligning them with human values becomes paramount. Reinforcement Learning from Human Feedback (RLHF) has emerged as a key technique, translating preference data into reward models when oracle human values remain inaccessible. In practice, RLHF mostly relies on approximate reward models, which may not consistently guide the policy toward maximizing the underlying human values. We propose Policy-Interpolated Learning for Aligned Feedback (PILAF), a novel response sampling strategy for preference labeling that explicitly aligns preference learning with maximizing the underlying oracle reward. PILAF is theoretically grounded, demonstrating optimality from both an optimization and a statistical perspective. The method is straightforward to implement and demonstrates strong performance in iterative and online RLHF settings where feedback curation is critical.

## 1. Introduction

Reinforcement Learning from Human Feedback (RLHF) (Ouyang et al., 2022) has revolutionized large language models (LLMs) by incorporating human preferences, enabling significant progress in applications such as conversational AI (Achiam et al., 2023), personalized tutoring (Limo et al., 2023), and content curation (Yue et al., 2024). At the core of RLHF is *reward modeling*, a critical process that translates human feedback—such as pairwise comparisons or rankings—into a measurable objective for model training. By formalizing human preferences, reward models then guide LLMs towards alignment through *policy optimization*.

While numerous studies have focused on improving language models (LMs) by optimizing fixed reward functions

(Dong et al., 2023; Liu et al., 2024b) or leveraging pre-existing preference datasets (Ethayarajh et al., 2024; Azar et al., 2024; Xu et al., 2024), comparatively less attention has been paid to the critical challenge of collecting *effective* data for human-labeling in the RLHF pipeline, to maximize its utility. This is an important problem, as the quality of preference data directly impacts the effectiveness of reward modeling and, consequently, the overall success of fine-tuning. This challenge is further compounded by the high cost of expert preference labeling (Lightman et al., 2023).

Preference data is usually generated by sampling response pairs $(\vec{y}_i^a, \vec{y}_i^b)$ to a prompt $x_i$ from a policy, and presenting them to human labelers for preference annotation. It is commonly assumed that the annotation follows the Bradley-Terry (BT) model, under an *oracle reward*. Next, we use maximum likelihood estimation (MLE) on these preference data to train a reward model, which then serves as a measurable objective to optimize the policy (i.e. LLM) while staying close to a reference policy. In Direct Preference Optimization (DPO) (Rafailov et al., 2023), this pipeline is simplified by optimizing the policy with implicit reward modeling. However, all these pipelines give rise to a *misalignment of objectives:* RLHF (or DPO) should, in principle, train its policy to maximize the (inherently inaccessible) *oracle objective* which combines the *oracle reward* from the BT model with reference regularization. In practice, RLHF relies on preference data through the MLE objective in reward modeling or through methods like DPO, which are *not* designed to guide policy optimization towards maximizing oracle rewards. Thus, reward optimization (either directly or implicitly via DPO) and (optimal) policy optimization are not inherently aligned, potentially leading to inefficiencies (see Section 2).

In this work, we study this misalignment by examining the sampling scheme that generates response pairs $(\vec{y}_i^a, \vec{y}_i^b)$ for preference labeling, which is especially important when additional preference data is collected mid-RLHF training to mitigate the off-policy distributional shift, as is empirically standard (Touvron et al., 2023; Bai et al., 2022). We show that uniform sampling from the current policy, as is common, leads to misaligned gradients of the two objectives (reward model loss and true oracle objective).

To tackle this issue, we present *Theoretically Grounded*

[1]New York University [2]Meta FAIR *Joint second authors [◇]Joint senior authors. Correspondence to: Yunzhen Feng <yf2231@nyu.edu>.

*Proceedings of the 42$^{nd}$ International Conference on Machine Learning*, Vancouver, Canada. PMLR 267, 2025. Copyright 2025 by the author(s).

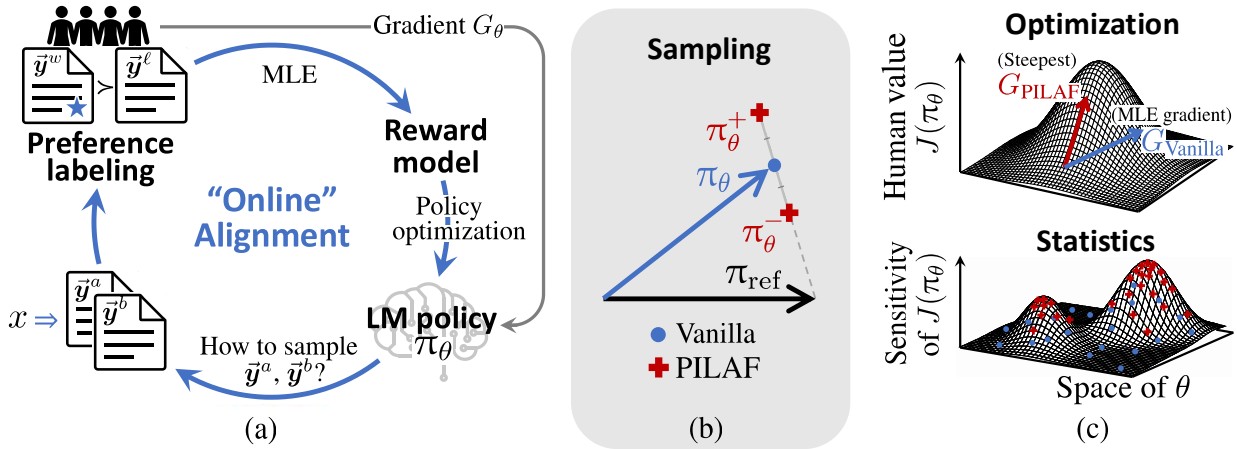

*Figure 1.* **Overview of our approach**. (a) We consider a full RLHF training setup, where a language model (LM) policy is iteratively refined through active data collection. Our goal is to develop an optimal response sampling method for preference labeling. (b) We introduce PILAF, which generates responses by interpolating between the current policy and a reference policy, balancing exploration and exploitation. (c) Our theoretical analysis shows that T-PILAF aligns the parameter gradient with the steepest direction for maximizing human values and achieves more favorable convergence in regions of high sensitivity.

*Policy-Interpolated Learning for Aligned Feedback* (T-PILAF), a novel sampling method that aligns reward modeling with value optimization. Specfically, T-PILAF generates responses by interpolating the policy model and the reference model for a balanced exploration and exploitation. We provide rigorous theoretical analysis to show that for preference data generated with T-PILAF, the gradient of the MLE loss with respect to the policy network's parameters is aligned with the policy gradient of the oracle objective in a first-order sense. This alignment enables the policy to optimize directly for the oracle value, achieving both alignment and efficiency. Furthermore, we separately show from a statistical perspective that T-PILAF aligns optimization with the steepest directions of the oracle objective. It thus makes the sampled preference pairs more informative, reducing variance and improving training stability.

We then present PILAF, a simple modification of our theoretical sampling scheme T-PILAF, which naturally lends itself to practical implementation. For clarity of exposition, we present our method in the context of DPO; however, PILAF can be adapted to a wide class of preference optimization methods.[1] See Figure 1 for an illustration of our setup, method, and the optimization and statistical principles underlying PILAF.

We conduct extensive experiments to validate PILAF's effectiveness and robustness. As a stand-in for expensive human annotators, we use a well-trained reward model—Skywork-Llama-3.1-8B (Liu et al., 2024a)—as a proxy for the oracle reward. Throughout training, we query this model exclusively for preference labels, simulating human feedback.

We then align the Llama-3.1-8B base model (Dubey et al., 2024) using these proxy-labeled preference data in two settings: iterative DPO (Xiong et al., 2024) and online DPO (Guo et al., 2024). In both scenarios, preference data is collected on-the-fly, either after each full training epoch in the iterative setting or after every training step in the online setting. Across all configurations, PILAF outperforms all the baselines, producing a policy with higher reward (as measured by the proxy) and a lower KL divergence from the reference model, reducing annotation and computation costs by over 40% in iterative DPO.

Our key contributions are as follows:

- *(Practical sampling algorithm)* We propose PILAF (Section 5), an efficient sampling algorithm for generating response pairs in the RLHF pipeline for improved sample efficiency and performance, derived from its theoretically grounded variant T-PILAF (Section 3).
- *(Theoretical optimality)* We provide theoretical guarantees for the efficiency of our approach from both optimization and statistical perspectives (Section 4).
- *(Empirical validation)* We validate PILAF in both iterative and online DPO settings (Section 6) and observe that it consistently outperforms baselines by achieving higher reward and lower KL divergence from the reference model. Moreover, PILAF achieves comparable performance at significantly reduced annotation and computational costs.

### 1.1. Related Work

**Existing Sampling Schemes.** In academic papers, uniform vanilla sampling is the most commonly used approach,

---

[1]See Appendix G for the extension to PPO.

while methods such as best-of-N and worst-of-N have also been explored (Dong et al., 2024). Xie et al. (2024) propose sampling one response from the current policy model and another from a reference model, modifying the loss function to encourage optimistic behavior. Similarly, Zhang et al. (2024) sample one response from the current model but rank it alongside two offline responses from the reference model. Shi et al. (2024) present a formula similar to ours based on intuition, introducing several hyperparameters and analyzing convergence speed with DPO in a tabular setting. Liu et al. (2024c) train an ensemble of reward models to approximate a posterior distribution over possible rewards and use Thompson sampling to generate responses with exploration. In contrast to these works, we theoretically establish the principles of response generation for preference labeling, making minimal assumptions and simplifications while demonstrating the optimality of our approach. Our approach eliminates the need for hyperparameter tuning.

**Policy Gradient.** Our theoretical principle is closely related to the family of policy gradient methods (Williams, 1992; Sutton et al., 1999) in reinforcement learning, which optimize a policy $\pi_\theta$ by estimating and ascending the gradient of the expected return $\nabla_\theta J(\theta)$. Significant advancements have been made to improve the efficiency of these methods, including variance reduction techniques (Greensmith et al., 2004), off-policy gradient estimation (Degris et al., 2012), interpolating on-policy and off-policy updates (Gu et al., 2017), deterministic policy gradients (Silver et al., 2014), and three-way robust estimation approaches (Kallus & Uehara, 2020). Our study extends these principles to preference learning for LMs, aligning the MLE gradient with the oracle objective gradient by controlling the response sampling distribution, thereby improving learning efficiency.

A review of other RLHF literature, particularly on data selection for the preference dataset, is deferred to Appendix A.

## 2. Problem Setup and Motivation

### 2.1. Aligning LMs with Human Preferences

**Language Model (LM)**. At the core of our framework is a language model that processes prompts $x \in \mathcal{X}$ and generates responses $\vec{y} \in \mathcal{Y}$. Each response is represented as a sequence of tokens $\vec{y} = (y_1, y_2, \ldots, y_T)$. The primary goal of RLHF is to guide the model to generate responses that align with human preferences. This translates to designing a decision policy $\pi$ (parameterized as a LM) that maps prompts to responses, maximizing a reward that reflects human preferences (with a KL regularization).

**Preference Data**. The oracle reward for human values is inherently inaccessible. Instead, the alignment process approximates the reward using a dataset of human-labeled preferences,

$$\mathcal{D} = \left\{ (x_i, \vec{y}_i^w, \vec{y}_i^\ell) \right\}_{i=1}^n,$$

where each sample contains: (i) a prompt $x_i$, independently drawn from a distribution $\rho$, and (ii) a pair of responses $(\vec{y}_i^w, \vec{y}_i^\ell)$, where $\vec{y}_i^w$ is preferred over $\vec{y}_i^\ell$ in human labeling. The response pair $(\vec{y}_i^w, \vec{y}_i^\ell)$ is first generated from a joint distribution $\mu(\cdot \mid x)$ and then presented to human labelers for preference annotation. Human preferences are commonly modeled using the *Bradley–Terry (BT)* model, which assumes:

$$\mathbb{P}\left(\vec{y}^a \succ \vec{y}^b \mid x\right) = \sigma\left(r^\star(x, \vec{y}^a) - r^\star(x, \vec{y}^b)\right), \quad (1)$$

where $r^\star(x, \vec{y})$ represents the (unknown) oracle reward of a response given a prompt, and $\sigma(z) = \{1 + \exp(-z)\}^{-1}$ is the sigmoid function, mapping differences in rewards to probabilities. We adopt the BT model throughout this paper.

**Reward Modeling**. The preference data, encoding human judgment, is then used to train a reward model, $r_\theta$, which serves as a measurable objective for training the policy model. $r_\theta$ is trained by solving a MLE objective:

$$\min_\theta \; \widehat{\mathcal{L}}(\theta) := -\frac{1}{n} \sum_{i=1}^n \log \sigma\left(r_\theta(x_i, \vec{y}_i^w) - r_\theta(x_i, \vec{y}_i^\ell)\right).$$
$$(2)$$

This empirical loss approximates the expected negative log-likelihood $\mathcal{L}(\theta) :=$

$$\mathbb{E}_{\substack{x \sim \rho \\ (\vec{y}^a, \vec{y}^b) \sim \mu(\cdot \mid x)}} \left[ -\log \sigma\left(r_\theta(x, \vec{y}^w) - r_\theta(x, \vec{y}^\ell)\right) \right]. \quad (3)$$

**Policy Optimization**. To align a language model $\phi$ with human preferences, we optimize it to maximize the learned rewards $r_\theta$ while staying close to a reference policy $\pi_{\text{ref}}$. The objective is

$$\max_\phi \; \mathbb{E}_{x \sim \rho, \vec{y} \sim \pi_\phi(\cdot \mid x)} \left[ r_\theta(x, \vec{y}) \right] - \beta D_{\text{KL}}(\pi_\phi \parallel \pi_{\text{ref}}). \quad (4)$$

It consists of two parts:

(i) The *reward* term $\mathbb{E}_{x \sim \rho, \vec{y} \sim \pi(\cdot \mid x)}[r_\theta(x, \vec{y})]$ encourages the policy to generate high-quality responses.

(ii) The *regularization* term $D_{\text{KL}}(\pi \parallel \pi_{\text{ref}})$ penalizes deviations from the reference policy $\pi_{\text{ref}}$ and is defined as $\mathbb{E}_{x \sim \rho}\left[D_{\text{KL}}(\pi(\cdot \mid x) \parallel \pi_{\text{ref}}(\cdot \mid x))\right]$.

Here, $\beta$ is a regularization parameter that balances the trade-off between reward maximization and adherence to the reference policy. We assume $\beta$ is fixed and practitioner-specified.

### 2.2. Direct Preference Optimization

The above-described RLHF pipeline typically leverages the Proximal Policy Optimization (PPO) algorithm (Schulman et al., 2017) to perform policy optimization. This approach requires loading the policy network, reward model, reference model, and a value model onto the GPU during training,

making it highly resource-intensive. To improve computational efficiency and practicality, Direct Preference Optimization (DPO) (Rafailov et al., 2023) has been proposed, enabling direct alignment without the need for a reward model or a value model.

A key insight of DPO is that any policy $\pi_\theta$ can be viewed as the optimal solution to problem (4) if the reward $r_\theta$ is

$$r_\theta(x, \vec{y}) := \beta \cdot \log \left( \frac{\pi_\theta(\vec{y} \mid x)}{\pi_{\text{ref}}(\vec{y} \mid x)} \right). \quad (5)$$

Thus, DPO can directly optimize the policy $\pi_\theta$ using $\widehat{\mathcal{L}}(\theta)$ in Equation (2), where $r_\theta$ is replaced by $\pi_\theta$ as defined in Equation (5). This reformulation makes the objective dependent solely on $\theta$, with the reward being implicitly learned through the policy itself. As a result, the optimization process becomes significantly more efficient.

### 2.3. Motivation: Realigning Oracle Reward Maximization

To fully align with human values, RLHF should, in principle, train the policy to maximize the oracle reward, $r^\star$, as defined in the BT model. The corresponding oracle objective is then:

$$J(\pi) := \mathbb{E}_{x \sim \rho, \, \vec{y} \sim \pi(\cdot|x)} \left[ r^\star(x, \vec{y}) \right] - \beta \, D_{\text{KL}}(\pi \parallel \pi_{\text{ref}}). \quad (6)$$

Since direct access to $r^\star$ is unavailable, RLHF instead relies on preference data, either through MLE-based reward modeling or methods like DPO. However, these processes are not inherently designed to train the policy to directly maximize the oracle objective, $J(\pi)$. The following comparison of the gradient will highlight the differences.

To make the notation concise, we introduce the following shorthands: $\Delta r^\star := r^\star(x, \vec{y}^a) - r^\star(x, \vec{y}^b)$, $\Delta r_\theta := r_\theta(x, \vec{y}^a) - r_\theta(x, \vec{y}^b)$ and $\boldsymbol{g} := \nabla_\theta r_\theta(x, \vec{y}^a) - \nabla_\theta r_\theta(x, \vec{y}^b)$. The lemmas give the following expressions for the gradients of the scalar value $J(\pi_\theta)$ (the optimal gradient towards the oracle objective) and the training loss $\mathcal{L}(\theta)$ (the gradient actually used by DPO):

**Lemma 2.1** (Gradient of value $J(\pi_\theta)$). *For any $\pi_\theta$, the gradient of the expected value $J(\pi_\theta)$ satisfies*

$$\nabla_\theta J(\pi_\theta) = \frac{1}{2\beta} \mathbb{E}_{x \sim \rho; \, \vec{y}^a, \vec{y}^b \sim \pi_\theta(\cdot|x)} \left[ \{\Delta r^\star - \Delta r_\theta\} \cdot \boldsymbol{g} \right]. \quad (7)$$

**Lemma 2.2** (Gradient of the loss function $\mathcal{L}(\theta)$ for vanilla sampling). *For any $\pi_\theta$ and the vanilla response sampling scheme, the gradient of the negative log-likelihood function $\mathcal{L}(\theta)$ is given by*

$$\nabla_\theta \mathcal{L}(\theta) = - \mathbb{E}_{x \sim \rho; \, \vec{y}^a, \vec{y}^b \sim \pi_\theta(\cdot|x)} \left[ \{\sigma(\Delta r^\star) - \sigma(\Delta r_\theta)\} \cdot \boldsymbol{g} \right]. \quad (8)$$

We observe that these two gradients share a similar structure. The key difference is $\Delta r^\star - \Delta r_\theta$ for $\nabla_\theta J(\pi_\theta)$ and $\sigma(\Delta r^\star) - \sigma(\Delta r_\theta)$ for $\nabla_\theta \mathcal{L}(\theta)$.

In this work, we design a sampling distribution $\mu$ to correct this mismatch. Our new sampling method is optimal in the sense that DPO, when using our sampling, will maximize the oracle objective $J(\pi)$, even without direct access to it. The sampling strategy improves the quality of the preference dataset, maximizes the utility of limited data, and enhances both performance and efficiency.

This focus is particularly crucial in scenarios where additional data is collected during mid-training—a key phase in the iterative fine-tuning of LMs (Touvron et al., 2023; Bai et al., 2022; Xiong et al., 2024; Guo et al., 2024). At this stage, a preliminary policy $\pi_\theta$ (distinct from $\pi_{\text{ref}}$) is already in place, but its performance may fall short of expectations. It is thus necessary to gather additional preference data, ideally on-policy data that target areas where the current policy shows room for improvement. An effective sampling design can significantly enhance the efficiency of leveraging human feedback in this process.

## 3. T-PILAF: Theoretical Sampling Scheme

We now present T-PILAF - *theoretically grounded policy interpolation for aligned feedback* - our sampling scheme for generating responses in data collection[2]. The scheme is shown (in Section 4) to be optimal from both optimization and statistical perspectives.

Consider we have an initial policy $\pi_\theta$ and aim to collect preference data to further refine its performance. We propose two complementary variants of policy $\pi_\theta$: one that encourages exploration in regions more preferred by $\pi_\theta$, reflecting an optimistic perspective, and another that focuses on areas less favored by $\pi_\theta$, reflecting a conservative adjustment.

Specifically, we define policies $\pi_\theta^+$ and $\pi_\theta^-$ around $\pi_\theta$ as

$$\pi_\theta^+(\vec{y} \mid x) := \frac{1}{Z_\theta^+(x)} \pi_\theta(\vec{y} \mid x) \exp\left\{ r_\theta(x, \vec{y}) \right\}, \quad (9a)$$

$$\pi_\theta^-(\vec{y} \mid x) := \frac{1}{Z_\theta^-(x)} \pi_\theta(\vec{y} \mid x) \exp\left\{ - r_\theta(x, \vec{y}) \right\}, \quad (9b)$$

where the reward function $r_\theta$ is defined in equation (5). The partition function $Z_\theta^+(x)$ (or $Z_\theta^-(x)$) is given by $Z_\theta^+(x) := \int_{\mathcal{Y}} \pi_\theta(\vec{y} \mid x) \exp\{r_\theta(x, \vec{y})\} \, d\vec{y}$.

For any prompt $x \in \mathcal{X}$, our sampling procedure involves the following steps:

(i) Draw a random variable $\xi$ from Bernoulli$(p_0(x))$, where $p_0(x) := Z_\theta^+(x) Z_\theta^-(x) / \{1 + Z_\theta^+(x) Z_\theta^-(x)\}$.

---

[2]The T in T-PILAF serves to distinguish the theoretical scheme from the derived, simplified, efficiently implementable PILAF.

(ii) If $\xi = 1$, independently draw responses $\vec{\boldsymbol{y}}^a, \vec{\boldsymbol{y}}^b \in \mathcal{Y}$ according to $\vec{\boldsymbol{y}}^a \sim \pi_\theta^+(\cdot \mid x)$ and $\vec{\boldsymbol{y}}^b \sim \pi_\theta^-(\cdot \mid x)$. If $\xi = 0$, draw responses as $\vec{\boldsymbol{y}}^a, \vec{\boldsymbol{y}}^b \sim \pi_\theta(\cdot \mid x)$.

In the next section, we will theoretically analyze T-PILAF. To account for the changes in sampling, we adopt a slightly modified loss function in the theoretical framework:

$$\widehat{\mathcal{L}}(\theta) := -\frac{1}{n}\sum_{i=1}^{n} w(x_i) \cdot \log \sigma\Big(r_\theta\big(x_i, \vec{\boldsymbol{y}}_i^w\big) - r_\theta\big(x_i, \vec{\boldsymbol{y}}_i^\ell\big)\Big).$$

The newly introduced weight function $w$ is defined as

$$w(x) := \big\{1 + Z_\theta^+(x)\, Z_\theta^-(x)\big\}/\,\overline{Z}_\theta\,, \qquad (10)$$

where the normalization constant $\overline{Z}_\theta > 0$ is given by $\overline{Z}_\theta := 1 + \int_{\mathcal{X}} Z_\theta^+(x)\, Z_\theta^-(x)\, \rho(x)\, dx$. We also modify the population loss $\mathcal{L}$ in Equation (3) with the weight function.

# 4. Theoretical Analysis

This section provides the theoretical grounding and analysis of our proposed sampling scheme from two perspectives. In the *optimization* analysis (Section 4.1) we show that T-PILAF *aligns two objectives (gradient alignment property)*: maximizing the likelihood function (Equation (3)) becomes equivalent to gradient ascent on the value function $J(\pi_\theta)$ (Equation (6)). Consequently, policy updates on $\pi_\theta$ move the parameters in the direction of steepest increase of $J$. T-PILAF thus provides the potential to accelerate training and improve generalization, compared to vanilla (uniform) sampling. In the *statistical* analysis (Section 4.2) we focus on statistical error and show that the asymptotic covariance of the estimated parameter $\widehat{\theta}$ (inversely) aligns with the Hessian of the objective function $J$ when sampling with T-PILAF. As a result, T-PILAF makes the sampled comparisons more informative, as they align with directions where $J$ is most sensitive. The net outcome is reduced statistical variance of our method through tighter concentration of estimates in directions that matter most for performance.

## 4.1. Optimization Considerations

We begin by analyzing the DPO algorithm from an optimization perspective.

Theorem 4.1 below formally illustrates how T-PILAF ensures alignment between the MLE gradient, $\nabla_\theta \mathcal{L}(\theta)$, and the oracle objective gradient, $\nabla_\theta J(\pi_\theta)$.

**Theorem 4.1** (Gradient structure in DPO training). *Using data collected from our proposed response sampling scheme T-PILAF, the gradient of $\mathcal{L}(\theta)$ satisfies*

$$\nabla_\theta \mathcal{L}(\theta) = -\frac{\beta}{\overline{Z}_\theta}\, \nabla_\theta J(\pi_\theta) + T_2\,,$$

*where the constant $\overline{Z}_\theta$ is defined in equation (10), and the*

*term $T_2$ represents a second-order error.*

The detailed proof of Theorem 4.1 is deferred to Appendix C.1. Recall from Lemma 2.1 and Lemma 2.2 that the difference between two gradients is the sigmoid function; the most notable technical contribution here is showing how to leverage our sampling scheme to approximate the derivative $\sigma'$ of the sigmoid function. By using T-PILAF sampling, we can transform the difference term of the form $\sigma(\Delta r^\star) - \sigma(\Delta r_\theta)$ in $\nabla_\theta \mathcal{L}(\theta)$ into a linear difference $\Delta r^\star - \Delta r_\theta$ in $\nabla_\theta J(\pi_\theta)$. This bridges the gap between the non-linear sigmoid differences and the linear reward differences.

Theorem 4.1 establishes the *gradient alignment* property, demonstrating that minimizing the likelihood-based loss function $\mathcal{L}$ closely aligns with maximizing the oracle objective function $J$, with only a minor second-order error. It highlights how the proposed sampling scheme enables the DPO framework to effectively guide the policy toward optimizing the expected reward. Beyond DPO, in Appendix G, we show how the same principle can be applied to PPO-based RLHF algorithms to help improve the sampling.

## 4.2. Statistical Considerations

From a statistical standpoint, we first examine the asymptotic distribution of the estimated parameter $\widehat{\theta}$ when it (approximately) solves the optimization problem (2). In Theorem 4.2, we formally characterize the randomness or statistical error inherent in $\widehat{\theta}$ under this idealized scenario. The detailed proof of Theorem 4.2 is provided in Appendix C.2.2.

**Theorem 4.2.** *Assume the reward model $r^\star$ in the BT model (1) satisfies $r^\star = r_{\theta^\star}$ for some parameter $\theta^\star$. Under mild regularity conditions, the estimate $\widehat{\theta}$ asymptotically follows a Gaussian distribution*

$$\sqrt{n}\,(\widehat{\theta} - \theta^\star) \xrightarrow{d} \mathcal{N}(\mathbf{0}, \boldsymbol{\Omega}) \qquad \text{as } n \to \infty\,.$$

*We have an estimate of the covariance matrix $\boldsymbol{\Omega}$:*

$$\boldsymbol{\Omega} \preceq C_1 \cdot \boldsymbol{\Sigma}_\star^{-1}\,,$$

*where $C_1 > 0$ is a universal constant. When using T-PILAF, the matrix $\boldsymbol{\Sigma}_\star$ is given by*

$$\boldsymbol{\Sigma}_\star := \mathbb{E}_{x\sim\rho}\Big[\mathrm{Cov}_{\vec{\boldsymbol{y}}\sim\pi^\star(\cdot|x)}\big[\nabla_\theta r^\star(x, \vec{\boldsymbol{y}}) \mid x\big]\Big]. \quad (11)$$

Next we analyze the performance of the output policy $\widehat{\pi} = \pi_{\widehat{\theta}}$ from Theorem 4.2 in terms of the expected value $J(\pi)$. In Theorem 4.3, we show that our proposed sampling method guarantees that the covariance of the statistical error in $\widehat{\theta}$ aligns inversely with the Hessian of $J$ at the optimal policy $\pi^\star$. This alignment prioritizes convergence efficiency along directions where the Hessian has large eigenvalues, adapting to the geometry of the optimiza-

tion landscape. It highlights the efficiency of our sampling scheme in reducing statistical error. For the detailed proof of Theorem 4.3, see Appendix C.2.3.

**Theorem 4.3.** *The value function $J(\pi)$ we define in equation* (6) *satisfies* $\nabla_\theta J(\pi^\star) = \mathbf{0}$ *and*

$$\nabla_\theta^2 J(\pi^\star) \;=\; -\frac{1}{\beta}\,\boldsymbol{\Sigma}_\star \tag{12}$$

*for matrix $\boldsymbol{\Sigma}_\star$ defined in equation* (11). *As a corollary, suppose $\boldsymbol{\Sigma}_\star$ is nonsingular, then there exists a constant $C_2 > 0$ such that for any $\varepsilon > 0$,*

$$\limsup_{n\to\infty} \mathbb{P}\left\{ J(\widehat{\pi}) < J(\pi^\star) - C_2 \cdot \frac{d\,(1+\varepsilon)}{n} \right\} \tag{13}$$

$$\leq\; \mathbb{P}\left\{ \chi_d^2 > (1+\varepsilon)\,d \right\} \leq \exp\left\{ -\frac{d}{2}\big(\varepsilon - \log(1+\varepsilon)\big) \right\}.$$

Our proposed sampling distribution μ ensures that the output policy $\widehat{\pi}$ performs predictably and reliably. The value gap $J(\pi^\star) - J(\widehat{\pi})$ asymptotically follows a chi-square distribution, irrespective of the problem instance details, such as the underlying reward model $r^\star$. This *structure-invariant statistical efficiency* allows the method to achieve asymptotically efficient estimates without requiring explicit knowledge of the model structure.

We further derive a general lemma describing how μ affect the covariance in Appendix B. This result provides broader insights into what constitutes good preference data in RLHF.

## 5. PILAF Algorithm

We now demonstrate that the T-PILAF sampling scheme defined in Equation (9a) and (9b) can be naturally extended into an efficient empirical algorithm (PILAF).

The first challenge in implementing these definitions lies in calculating the normalizing factors $Z_\theta^+(x)$ and $Z_\theta^-(x)$, which can be computationally expensive for LLMs. To address this, we simplify the process by omitting these factors and replacing them with 1.[3] Consequently, the sampling process becomes straightforward: with probability $1/2$, we sample using $\pi_\theta$, and otherwise, we sample using $\pi_\theta^+$ and $\pi_\theta^-$.

The second challenge lies in sampling a response $\vec{y}$ from $\pi_\theta(\vec{y} \mid x) \exp\left\{ \pm r_\theta(x, \vec{y}) \right\}$ in an autoregressive way for next-token generation. We argue that the policy $\pi_\theta^+$ (and $\pi_\theta^-$) can be approximated in a token-wise manner:

$$\pi_\theta^+(\vec{y} \mid x) \approx \pi_\theta^+(y_1 \mid x)\,\pi_\theta^+(y_2 \mid x, y_1)$$
$$\cdots \pi_\theta^+(y_t \mid x, y_{1:t-1}) \cdots \pi_\theta^+(y_T \mid x, y_{1:T-1}),$$

---

[3]When the regularization coefficient $\beta$ is sufficiently small, the term $\exp\{r_\theta(x, \vec{y})\}$ in equation (9a) stays close to 1 and has only a minor effect. Consequently, the partition function $Z_\theta^+(x)$ is approximately 1. A similar reasoning applies to $Z_\theta^-(x)$.

where $\pi_\theta^+(y_t \mid x, y_{1:t-1}) =$

$$\frac{1}{Z(x, y_{1:t-1})} \pi_\theta(y_t \mid x, y_{1:t-1})\left( \frac{\pi_\theta(y_t \mid x, y_{1:t-1})}{\pi_{\text{ref}}(y_t \mid x, y_{1:t-1})} \right)^\beta$$

with $Z(x, y_{1:t-1})$ being a partition function. The substitution of $r_\theta$ uses the correspondence between the reward model $r_\theta$ and the policy $\pi_\theta$ in Equation (5), under the assumption that this correspondence holds for all truncations $y_{1:t-1}$. It gives us a direct per-token prediction rule:

$$\pi_\theta^+(\cdot \mid x, y_{1:t-1})$$
$$= \text{softmax}\Big( \big\{ (1+\beta)\,\boldsymbol{h}_\theta - \beta\,\boldsymbol{h}_{\text{ref}} \big\}(x, y_{1:t-1}) \Big).$$

Here $\boldsymbol{h}_\theta$ and $\boldsymbol{h}_{\text{ref}}$ are the logits of the policies $\pi_\theta$ and $\pi_{\text{ref}}$, respectively. $\beta$ is the regularization coefficient from the objective function $J(\pi)$ in Equation (6). Responses are then generated using standard decoding techniques, such as greedy decoding or nucleus sampling. Similarly, the generation for $\pi_\theta^-$ follows

$$\pi_\theta^-(\cdot \mid x, y_{1:t-1})$$
$$= \text{softmax}\Big( \big\{ (1-\beta)\,\boldsymbol{h}_\theta + \beta\,\boldsymbol{h}_{\text{ref}} \big\}(x, y_{1:t-1}) \Big).$$

For a detailed, step-by-step proof, see Appendix D.4.

We formalize our final algorithm in Algorithm 1. Vanilla DPO (Rafailov et al., 2023; Guo et al., 2024) employs a basic generation approach, sampling $\vec{y}_i^a, \vec{y}_i^b \sim \pi_\theta$ at Step 3. In contrast, instead of only sampling from $\pi_\theta$, our sampling scheme interpolates and extrapolates the logits $\boldsymbol{h}_\theta$ and $\boldsymbol{h}_{\text{ref}}$ with coefficient $\beta$, enabling exploration of a wider response space to align learning from human preference with value optimization. The $\beta$ here is the same parameter that controls the KL regularization in Equation (4), as set by the problem.

**Cost analysis.** We summarize sampling and annotation costs per preference pair for PILAF and related sampling schemes in Table 1. In *Vanilla* sampling (from $\pi_\theta$), two generations and two annotations are required for human preference labeling, same to PILAF when the pair is sampled from $\pi_\theta$, which happens half the time. With 50% probability, PILAF uses $\pi_\theta^+$ and $\pi_\theta^-$ to generate, requiring two forward passes with $\pi_\theta$ and $\pi_{\text{ref}}$ to generate one sample. Thus, on average, a preference pair sampled with PILAF requires a sampling cost of 3 forward passes (1.5 time the cost of *Vanilla*) with the same annotation cost. To compare, Xiong et al. (2024); Dong et al. (2024) perform *Best-of-N* sampling with $N = 8$, which generates and annotates all 8 responses, selecting the best and worst of them. Xie et al. (2024) use a *Hybrid* method that generates with $\pi_\theta$ and $\pi_{\text{ref}}$, thus matching the sampling and annotation costs of the *Vanilla* method. We empirically compare PILAF with these methods in the next section.

*Table 1.* A cost summary of PILAF and sampling methods from related works. *Best-of-N* method in Xiong et al. (2024) uses the oracle reward to score all candidate responses, then selects the highest- and lowest-scoring ones—instead of providing a preference label for only two responses. We restrict the oracle to providing only preference labels. Thus, we create a *Best-of-N* variant that uses the DPO internal reward for selection and then applies preference labeling, with an annotation cost of 2. We compare with this variant in the experiment.

| METHOD | $\bar{\boldsymbol{y}}^a$ | $\bar{\boldsymbol{y}}^b$ | SAMPLING COST | ANNOTATION COST |
|---|---|---|---|---|
| *Vanilla* (RAFAILOV ET AL., 2023) | $\pi_\theta$ | $\pi_\theta$ | 2 | 2 |
| *Best-of-N* (XIONG ET AL., 2024), N=8 | BEST OF $\pi_\theta$ | WORST OF $\pi_\theta$ | 8 | 8* |
| *Best-of-N* (WITH DPO REWARD), N=8 | BEST OF $\pi_\theta$ | WORST OF $\pi_\theta$ | 8 | 2 |
| *Hybrid* (XIE ET AL., 2024) | $\pi_\theta$ | $\pi_{\text{ref}}$ | 2 | 2 |
| *PILAF* (OURS) | $\pi_\theta^+/\pi_\theta$ | $\pi_\theta^-/\pi_\theta$ | 3 | 2 |

---

**Algorithm 1** DPO with PILAF (ours).

**input** Prompt Dataset $\mathcal{D}_\rho$, preference oracle $\mathcal{O}$, $\pi_\theta$, $\pi_{\text{ref}}$.
1: **for** step $t = 1, ..., T$ **do**
2:     Sample $n_t$ prompts $\{x_i\}_{i=1}^{n_t}$ from $\mathcal{D}_\rho$.
3:     With probability 1/2, sample $\bar{\boldsymbol{y}}_i^a, \bar{\boldsymbol{y}}_i^b \sim \pi_\theta$; with probability 1/2, sample $\bar{\boldsymbol{y}}_i^a \sim \pi_\theta^+$ and $\bar{\boldsymbol{y}}_i^b \sim \pi_\theta^-$.
4:     Query $\mathcal{O}$ to label $(x_i, \bar{\boldsymbol{y}}_i^a, \bar{\boldsymbol{y}}_i^b)$ into $(x_i, \bar{\boldsymbol{y}}_i^w, \bar{\boldsymbol{y}}_i^\ell)$.
5:     Update $\pi_{\theta_t}$ with DPO loss using $\{(x_i, \bar{\boldsymbol{y}}_i^w, \bar{\boldsymbol{y}}_i^\ell)\}_{i=1}^{n_t}$.
6: **end for**

## 6. Experiments

In this section, we empirically evaluate PILAF in both an iterative DPO setting (Section 6.1, following Xiong et al. (2024); Dong et al. (2024)) and an online DPO setting (Section 6.2, following Guo et al. (2024)) where the model undergoes multiple rounds of refinement through active data collection. Our findings indicate that, without requiring any hyper-parameter tuning, our sampling scheme stabilizes training, achieves higher reward scores, and maintains lower KL divergence from the reference model.

**General Setup**. We align the Llama-3.1-8B base model (Dubey et al., 2024) in terms of helpfulness and harmlessness using the HH-RLHF dataset (Bai et al., 2022), a widely-used benchmark dataset for alignment. It consists of 161k prompts in the training set. For response preference labeling, we use a well-trained reward model to simulate human preferences by assigning preference to pairs of responses under the BT assumption in Equation (1). Specifically, we employ the Skywork-Reward-8B model (Liu et al., 2024a), a top-performing 8B model on RewardBench (Lambert et al., 2024), as our oracle $\mathcal{O}$. During training, interaction with this reward model is limited to providing two responses for comparison. We set $\beta = 0.1$ in all the experiments.

**Supervised Fine-Tuning (SFT)**. To initialize training, following Rafailov et al. (2023), we first fine-tune the base model to obtain the SFT model as $\pi_{\text{ref}}$, which we fix as the reference model in all the experiments. We use the originally preferred responses from the HH-RLHF dataset as the

SFT dataset and perform full-parameter tuning.

**Evaluation**. We present our results using the reward-KL curve, following Gao et al. (2023), with the reward evaluated by the oracle reward model $\mathcal{O}$. To monitor the impact of our sampling scheme on the optimization trajectory, we evaluate the model every 50 gradient steps during training. We use the entire testset of HH-RLHF (8.55K samples) to evaluate.

**Baselines**. We compare our sampling method against existing methods in Table 1, and with VPO (Cen et al., 2024), which uses the vanilla sampling but incorporates an explicit exploration term in the loss. Since we treat the oracle $\mathcal{O}$ as a proxy for human labelers that can only provide pairwise preferences, all baselines are constrained to query the oracle with exactly two samples at a time. We thus adapt a *Best-of-N* variant that deploys the internal DPO reward to select the top and bottom candidates, which are then presented to the oracle for preference labeling, as listed in Table 1. We compare PILAF against the baselines: *Vanilla Sampling*, *Best-of-N Sampling* (with DPO reward), *Hybrid Sampling* combined with a modified DPO loss (Xie et al., 2024), and *VPO* (Cen et al., 2024).

Full experimental details can be found in Appendix F.

### 6.1. Iterative DPO

**Implementation**. We first consider the iterative DPO framework (Xiong et al., 2024; Dong et al., 2024), in which preference data is collected in successive iterations rather than as a single fixed dataset. At the start of each iteration, a large dataset of responses is sampled using the current model, annotated for preferences, and then used to train the current model. Concretely, we set $n_t = |\mathcal{D}_\rho|$ in Algorithm 1, meaning that all prompts are used to generate new responses at each iteration. During the first iteration, when $\pi_{\text{ref}}$ and $\pi_\theta$ are identical, PILAF reduces to *Vanilla Sampling*. Hence, we choose to focus our comparison on the second iteration. For consistency, we initialize all runs with the same policy model obtained at the end of the first iteration via *Vanilla Sampling*.

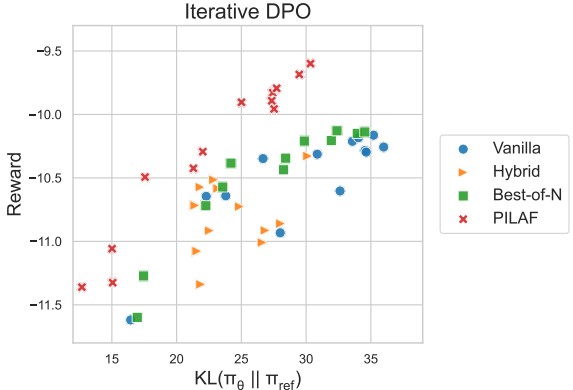

*Figure 2.* **Reward-KL curve for Iterative DPO**. All training runs start from the same model obtained at the end of the first iteration via *Vanilla Sampling*. Each dot represents an evaluation performed every 50 training steps.

*Table 2.* **Results of Iterative DPO**. We report the average reward, KL divergence from the reference model, and objective $J$ on the testset. Higher reward and $J$ are better, while lower KL divergence is better. We use **boldface** to indicate the best result and underline to denote the second-best result.

| METHOD | REWARD ($\uparrow$) | KL ($\downarrow$) | $J$ ($\uparrow$) |
|---|---|---|---|
| *Vanilla* | -10.16 | 35.20 | -13.68 |
| *Best-of-N* | -10.13 | 32.38 | -13.37 |
| *Hybrid* | -10.51 | **22.86** | -12.80 |
| *PILAF* (OURS) | **-9.80** | 25.01 | **-12.30** |

**Results**. Figure 2 presents the reward-KL curve for iterative DPO. PILAF significantly outperforms all the other methods: it achieves the end-point rewards of the baselines already around halfway through training, with around 40% less training time. This reduction directly translates to savings in both annotation and computational costs. We summarize the final performance in Table 2. PILAF produces a final policy with a high reward value and a modestly small KL divergence from the reference model, thereby achieving the highest overall objective $J$.

### 6.2. Online DPO

**Implementation.** We further evaluate our sampling method in the online DPO setting (Guo et al., 2024), where new responses are generated and labeled at every training step, and these preference data are immediately used to update $\pi_\theta$. This setting corresponds to the case where $n_t$ (in Algorithm 1) is set to the training batch size, resulting in the most annotation-intensive and most actively on-policy alignment. By collecting and utilizing preference data on the fly for each batch, the policy is continuously refined using on-policy feedback throughout the entire training process. Similar to Iterative DPO, we initialize all training runs with

the same $\pi_\theta$ and focus on comparing the subsequent optimization. Further details are in Appendix F.

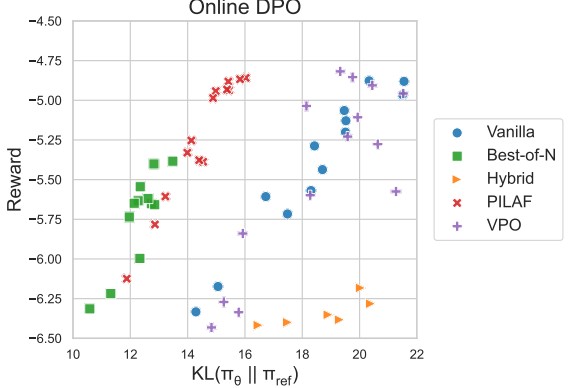

*Figure 3.* **Reward-KL curve for Online DPO**. Each dot represents an evaluation performed every 50 training steps.

**Results**. Figure 3 demonstrates the effectiveness of PILAF in the pure online setting, and we summarize the final performance in Table 3. Compared with *Vanilla*, *VPO*, and *Hybrid Sampling*, PILAF achieves a significantly better Reward-KL trade-off curve, attaining higher reward with lower KL. Although *Vanilla* and *VPO* eventually achieve roughly the same reward value as PILAF, it comes at the cost of a substantially higher KL. When compared with *Best-of-N*, PILAF traces a similar Reward–KL trajectory but ends with a higher reward and a better final objective after the same number of iterations, translating to lower sample complexity and reduced annotation and computational cost.

*Table 3.* **Results of Online DPO.** We report the average reward, KL divergence from the reference model, and objective $J$ on the testset.

| METHOD | REWARD ($\uparrow$) | KL ($\downarrow$) | $J$ ($\uparrow$) |
|---|---|---|---|
| *Vanilla* | -4.96 | 21.50 | -7.11 |
| *Best-of-N* | -5.54 | **12.35** | -6.77 |
| *Hybrid* | -6.42 | 16.46 | -8.96 |
| *VPO* | -4.91 | 22.31 | -7.09 |
| *PILAF* (OURS) | **-4.88** | 15.42 | **-6.42** |

**Robustness Analysis**. Having established the effectiveness of PILAF, we further evaluate its robustness by testing whether it improves optimization and statistical convergence under challenging conditions, as predicted from our statistical theory in Section 4.2. Specifically, we replace the initial model with one that has overfit on a fixed off-policy dataset. This setup allows us to examine how different methods handle optimization starting from a poor initial point.

In Figure 4, we compare the performance of PILAF and *Vanilla Sampling* when both are initialized from an overfitted policy. We observe that *Vanilla Sampling* rapidly

increases its KL divergence from the reference model while its reward improvement diminishes over time. In contrast, PILAF undergoes an early training phase with fluctuating KL values but ultimately attains a policy with higher reward and substantially lower KL divergence. We hypothesize that PILAF's interpolation-based exploration design enables it to escape the suboptimal region of the loss landscape in which *Vanilla* remains. These results underscore PILAF's effectiveness in more robustly optimizing overfitted (or even adversarially initialized) policies.

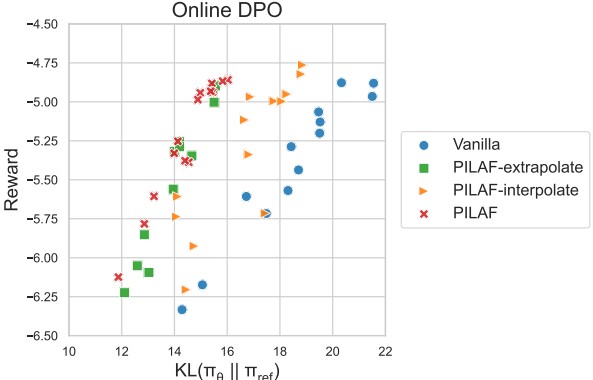

*Figure 5.* **Reward-KL curve for Online DPO with ablations**. Each dot represents an evaluation performed every 50 steps.

our theoretical prediction that the full PILAF algorithm is the best performing approach.

# 7. Conclusion

In this paper, we introduced Policy-Interpolated Learning for Aligned Feedback (PILAF), a novel sampling method designed to enhance response sampling for preference labeling. Theoretical analysis highlights PILAF's superiority from both optimization and statistical perspectives, demonstrating its ability to stabilize training, accelerate convergence, and reduce variance. The method is straightforward to implement and requires no additional hyperparameter tuning. We empirically validated its performance in both iterative DPO and online DPO settings, where it consistently outperformed existing approaches. To achieve the same level of performance, PILAF consistently requires lower annotation costs, which can be substantial when annotations require experts in knowledge-intensive domains.

In future work, we hope to extend PILAF to other paradigms, such as KTO (Ethayarajh et al., 2024) and IPO (Azar et al., 2024). Due to resource constraints, our evaluations were conducted using 8B models and a reward model to simulate human feedback. Future studies involving larger-scale experiments and real human labeling would further generalize our findings.

Overall, this work takes an important step toward improving preference data curation in RLHF pipelines, laying the groundwork for more effective methods in alignment.

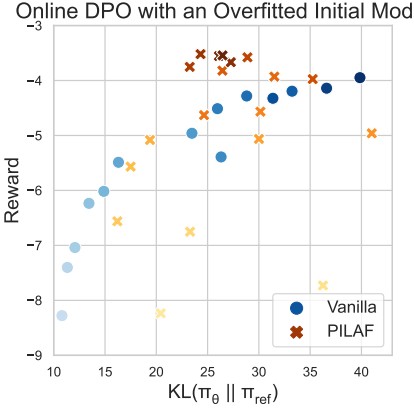

*Figure 4.* **Online DPO with an overfitted initial policy**. Each dot represents an evaluation performed every 50 training steps. Color saturation indicates the training step, with darker colors representing later steps.

## 6.3. Ablations

We further conduct two ablation studies to isolate the contributions of PILAF's interpolation and extrapolation components. Each component was replaced individually with vanilla sampling, yielding two baselines: one with $(\vec{y}^a, \vec{y}^b) = (\pi_\theta^+, \pi_\theta)$ (ablation of the interpolation component) and one with $(\vec{y}^a, \vec{y}^b) = (\pi_\theta, \pi_\theta^-)$ (ablation of the extrapolation component). We denote these ablation variants as PILAF-extrapolate and PILAF-interpolate, where one response is obtained via vanilla sampling and the other via extrapolation or interpolation, respectively.

The results are presented in Figure 5. Our theory suggests that the two sampling responses should come from different distributions in order to yield a controlled difference that the model can effectively learn from. Both ablation variants introduce such differences and outperform vanilla sampling. However, the variant with only interpolation (combined with vanilla sampling for the other response) performs much worse than full PILAF, highlighting the importance of the extrapolation response. The PILAF-extrapolate variant achieves slightly worse final results, and its convergence is much slower (each dot in our figure represents one evaluation after 50 steps). Overall, these ablation results confirm

## Acknowledgment

YF and JK acknowledge support through NSF NRT training grant award 1922658. YD acknowledges support from NSF grant DMS-2413812. The authors would like to thank Gabriel Synnaeve, He He, Wei Xiong, Pu Yang, Angelica Chen for helpful discussions. This work was supported in part through the NYU IT High Performance Computing resources, services, and staff expertise.

## Impact Statement

Our work aims to advance the state of Machine Learning—particularly in aligning large language models with active preference feedback. We believe that our work could contribute to more reliable and user-aligned language models, potentially improving downstream applications. By reducing the amount of annotation needed, the approach may lower the barriers to producing specialized models in resource-constrained environments. However, our experiments are currently limited to using a reward model to simulate human preferences. Additional studies with real human annotation are necessary to understand the risk of misalignment.

While our work has implications for various societal aspects, we do not identify any specific consequences to be specifically highlighted here.

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

# Contents

## A. Additional Literature Review

**RLHF**. RLHF has emerged as a cornerstone methodology for aligning large language models with human values and preferences (Achiam et al., 2023). Early systems (Ouyang et al., 2022) turn human preference data into reward modeling to optimize model behavior accordingly. DPO has been proposed as a more efficient approach that directly trains LLMs on preference data. As LLMs evolve during training, continuing training on pre-generated preference data becomes suboptimal due to the distribution shift. Empirically, RLHF is applied iteratively—generating on-policy data at successive stages to enhance alignment and performance (Touvron et al., 2023; Bai et al., 2022). Similarly, researchers have introduced iterative DPO (Xiong et al., 2024; Xu et al., 2023) and online DPO (Guo et al., 2024) to fully leverage online preference labeling. Ultimately, the quality of preference data play a critical role in determining the effectiveness of the alignment.

**Sampling in Frontier LLMs**. Technical reports of Frontier LLMs briefly mention sampling techniques. For instance,

Claude (Bai et al., 2022) utilizes models from different training steps to generate responses, while Llama-2 (Touvron et al., 2023) further use different temperatures for sampling. However, no further details are provided, leaving the development of a principled method an open challenge.

**Data Selection.** There is a line of research aimed at improving sample efficiency for preference labeling by selecting question and response pairs. Scheid et al. (2024) conceptualize this as a regret minimization problem, leveraging methods from linear dueling bandits. Das et al. (2024); Mehta et al. (2023); Muldrew et al. (2024); Ji et al. (2024) draw insights from active learning, using various uncertainty estimators to guide selection by prioritizing sample pairs with maximum uncertainty. These approaches focus directly on a dataset of questions and responses and are orthogonal to our work.

**Other Changes in Response Sampling.** Several works also modify the sampling design directly (Liu et al., 2024b; Dong et al., 2023), but with the goal of improving policy network optimization based on a reward model, rather than enhancing the reward modeling itself. Liu et al. (2024b) employ rejection sampling to approximate the response distribution induced by the reward model, thereby improving optimization. However, this approach requires access to the reward model and incurs higher computational and labeling costs. Similarly, Dong et al. (2023) use best-of-N sampling with the reward model to generate high-quality data for supervised fine-tuning (SFT). We consider these approaches orthogonal to our work.

Additionally, Cen et al. (2024) introduce a bonus term in the policy learning phase of online RLHF to promote exploration in response sampling, which aligns with the optimism principle.

## B. Additional Statistical Results

In addition to our analysis of T-PILAF in Section 3, here we present a generalized version of Theorem 4.2 that applies to any response sampling distribution $\mu$. While not directly tied to the main focus of this work, this broader result may be of independent interest to readers. The proof of Lemma B.1 is provided in Appendix C.2.1.

**Lemma B.1.** *For a general sampling distribution $\mu$, the statement in Theorem 4.2 remains valid with the matrix $\Sigma_\star$ redefined as*

$$\Sigma_\star := \mathbb{E}_{x\sim\rho,(\vec{\boldsymbol{y}}^a,\vec{\boldsymbol{y}}^b)\sim\overline{\mu}(\cdot|x)}\Big[\, w(x)\cdot\mathrm{Var}\big(\mathbb{1}\{\vec{\boldsymbol{y}}^a=\vec{\boldsymbol{y}}^w\}\mid x,\vec{\boldsymbol{y}}^a,\vec{\boldsymbol{y}}^b\big)\cdot\boldsymbol{g}\,\boldsymbol{g}^\top\Big], \tag{14}$$

*where the expectation is taken over the distribution*

$$\overline{\mu}(\vec{\boldsymbol{y}}^a,\vec{\boldsymbol{y}}^b\mid x):=\frac{1}{2}\left\{\mu(\vec{\boldsymbol{y}}^a,\vec{\boldsymbol{y}}^b\mid x)+\mu(\vec{\boldsymbol{y}}^b,\vec{\boldsymbol{y}}^a\mid x)\right\}. \tag{15a}$$

*The variance term is specified as*

$$\mathrm{Var}\big(\mathbb{1}\{\vec{\boldsymbol{y}}^a=\vec{\boldsymbol{y}}^w\}\mid x,\vec{\boldsymbol{y}}^a,\vec{\boldsymbol{y}}^b\big)=\sigma\big(r^\star(x,\vec{\boldsymbol{y}}^a)-r^\star(x,\vec{\boldsymbol{y}}^b)\big)\,\sigma\big(r^\star(x,\vec{\boldsymbol{y}}^b)-r^\star(x,\vec{\boldsymbol{y}}^a)\big) \tag{15b}$$

*and the gradient difference $\boldsymbol{g}$ is defined as*

$$\boldsymbol{g}:=\nabla_\theta\, r^\star(x,\vec{\boldsymbol{y}}^a)-\nabla_\theta\, r^\star(x,\vec{\boldsymbol{y}}^b). \tag{15c}$$

The general form of the matrix $\Sigma_\star$ offers valuable insights for designing a sampling scheme. To ensure $\Sigma_\star$ is well-conditioned (less singular), we must balance two key factors when selecting responses $\vec{\boldsymbol{y}}^a$ and $\vec{\boldsymbol{y}}^b$:

*Large variance:* The variance in definition (15b) should be maximized. This occurs when $r^\star(x,\vec{\boldsymbol{y}}^a)\approx r^\star(x,\vec{\boldsymbol{y}}^b)$. Intuitively, preference feedback is most informative when annotators compare responses of similar quality.

*Large gradient difference:* The gradient difference $\boldsymbol{g}$ from definition (15c) should also be large. This requires responses with significantly different gradients. Only then can the comparison provide a clear and meaningful direction for model training.

## C. Proof of Main Results

This section provides the proofs of the main results from Section 4, covering both optimization and statistical aspects. In Appendix C.1, we prove Theorem 4.1, which establishes the gradient alignment property. For the statistical results, Appendix C.2 begins with the proofs of Lemma B.1 and Theorem 4.2, which derive the asymptotic distribution of the estimated parameter $\widehat{\theta}$, and concludes with the proof of Theorem 4.3, analyzing the asymptotic behavior of the value gap $J(\pi^\star)-J(\widehat{\pi})$.

## C.1. Optimization Considerations: Proof of Theorem 4.1

We begin by presenting a rigorous restatement of Theorem 4.1, formally detailed in Theorem C.1 below.

**Theorem C.1** (Gradient structure in DPO training). *Consider the expected loss function $\mathcal{L}(\theta)$ during the DPO training phase. Using data collected from our poposed response sampling scheme* μ, *the gradient of $\mathcal{L}(\theta)$ satisfies*

$$\nabla_\theta \mathcal{L}(\theta) \;=\; -\frac{\beta}{\overline{Z}_\theta} \nabla_\theta J(\pi_\theta) \,+\, T_2 \,,$$

*where the constant $\overline{Z}_\theta$ is defined in equation* (10), *and the term $T_2$ represents a second-order error.*

*To control term $T_2$, assume the following uniform bounds:*

*(i) $\|r^\star\|_\infty \leq R$.*

*(ii) For any policy $\pi_\theta \in \Pi$, the induced reward $r_\theta$ satisfies $\|r_\theta\|_\infty \leq R$ and $\sup_{x,\vec{y}}\|\nabla_\theta r_\theta(x,\vec{y})\|_2 \leq G$.*

*Under these conditions, $T_2$ is bounded as*

$$\|T_2\|_2 \leq C \,\cdot\, \mathbb{E}_{x\sim\rho,\,\vec{y}^a,\vec{y}^b\sim\pi_\theta(\cdot|x)}\left[\left\{\left(r^\star(x,\vec{y}^a)-r^\star(x,\vec{y}^b)\right)-\left(r_\theta(x,\vec{y}^a)-r_\theta(x,\vec{y}^b)\right)\right\}^2\right],$$

*where the constant $C$ is given by $C = 0.1\,(1+e^{2R})\,G\big/\overline{Z}_\theta$.*

The proof of Theorem C.1 is structured into three sections. In Appendix C.1.1, we lay the foundation by presenting the key components, including the explicit expressions for the gradients $\nabla_\theta J(\pi_\theta)$ and $\nabla_\theta \mathcal{L}(\theta)$, as well as for the sampling density $\overline{\mu}$. Then Appendix C.1.2 establishes the connection between $\nabla_\theta J(\pi_\theta)$ and $\nabla_\theta \mathcal{L}(\theta)$ by leveraging these results, completing the proof of Theorem 4.1. Finally, in Appendix C.1.3, we provide a detailed derivation of the form of density function $\overline{\mu}$.

### C.1.1. BUILDING BLOCKS

To establish Theorem 4.1, which uncovers the relationship between the gradients of the expected value $J(\pi_\theta)$ and the negative log-likelihood function $\mathcal{L}(\theta)$, the first step is to derive explicit expressions for the gradients of both functions. The results are presented in Lemmas C.2 and C.3, with detailed proofs provided in Appendices D.1.2 and D.1.3, respectively.

**Lemma C.2** (Gradient of value $J(\pi_\theta)$). *For any $\pi_\theta$ in the parameterized policy class $\Pi$, the gradient of the expected value $J(\pi_\theta)$ satisfies*

$$\nabla_\theta J(\pi_\theta) \;=\; \frac{1}{2\beta} \mathbb{E}_{x\sim\rho;\,\vec{y}^a,\vec{y}^b\sim\pi_\theta(\cdot|x)}\left[\left\{\left(r^\star(x,\vec{y}^a)-r^\star(x,\vec{y}^b)\right)-\left(r_\theta(x,\vec{y}^a)-r_\theta(x,\vec{y}^b)\right)\right\}\right.$$
$$\left.\cdot\left\{\nabla_\theta r_\theta(x,\vec{y}^a)-\nabla_\theta r_\theta(x,\vec{y}^b)\right\}\right]. \quad (16)$$

**Lemma C.3** (Gradient of the loss function $\mathcal{L}(\theta)$). *For any $\pi_\theta$ in the parameterized policy class $\Pi$ and any sampling distribution* μ *of the responses, the gradient of the negative log-likelihood function $\mathcal{L}(\theta)$ is given by*

$$\nabla_\theta \mathcal{L}(\theta) \;=\; -\mathbb{E}_{x\sim\rho;\,(\vec{y}^a,\vec{y}^b)\sim\overline{\mu}(\cdot|x)}\left[w(x)\cdot\left\{\sigma\left(r^\star(x,\vec{y}^a)-r^\star(x,\vec{y}^b)\right)-\sigma\left(r_\theta(x,\vec{y}^a)-r_\theta(x,\vec{y}^b)\right)\right\}\right.$$
$$\left.\cdot\left\{\nabla_\theta r_\theta(x,\vec{y}^a)-\nabla_\theta r_\theta(x,\vec{y}^b)\right\}\right], \quad (17a)$$

*where the average density $\overline{\mu}$ is defined as*

$$\overline{\mu}(\vec{y}^a,\vec{y}^b\mid x) \;:=\; \frac{1}{2}\left\{\mu(\vec{y}^a,\vec{y}^b\mid x)+\mu(\vec{y}^b,\vec{y}^a\mid x)\right\} \quad (17b)$$

*as previously introduced in Equation* (15a).

In Lemma C.3, we observe that the gradient $\nabla_\theta \mathcal{L}(\theta)$ is expressed as an expectation over the probability distribution $\overline{\mu}$. By applying the sampling scheme outlined in Section 3, we can derive a more detailed representation of $\nabla_\theta \mathcal{L}(\theta)$. This refined form will reveal its close relationship to the gradient $\nabla_\theta J(\pi_\theta)$ given in expression (16).

Before moving forward, it is crucial for us to first derive the explicit form of $\overline{\mu}$. Specifically, we claim that the distribution $\overline{\mu}$ satisfies the following property

$$\frac{\overline{\mu}(\vec{y}^a, \vec{y}^b \mid x)}{\pi_\theta(\vec{y}^a \mid x)\,\pi_\theta(\vec{y}^b \mid x)} \;=\; \frac{1}{2\left\{1 + Z_\theta^+(x)\,Z_\theta^-(x)\right\}} \cdot \frac{1}{\sigma'\big(r_\theta(x,\vec{y}^a) - r_\theta(x,\vec{y}^b)\big)}\,, \tag{18}$$

where $\sigma'$ denotes the derivative of the sigmoid function $\sigma$, given by

$$\sigma'(z) \;=\; \frac{1}{(1+\exp(-z))(1+\exp(z))} \;=\; \sigma(z)\,\sigma(-z) \qquad \text{for any } z \in \mathbb{R}\,. \tag{19}$$

With these key components in place, we are now prepared to prove Theorem 4.1.

### C.1.2. DERIVATION OF THEOREM 4.1

With the tools provided by Lemmas C.2 and C.3 and the sampling density expression in (18), we are now ready to prove Theorem 4.1.

We begin by applying Lemma C.3 and reformulating equation (17a) as

$$\begin{aligned}
\nabla_\theta \mathcal{L}(\theta) \;=\; &- \mathbb{E}_{x\sim\rho;\; \vec{y}^a, \vec{y}^b\sim\pi_\theta(\cdot|x)} \Bigg[ w(x) \cdot \frac{\overline{\mu}(\vec{y}^a, \vec{y}^b \mid x)}{\pi_\theta(\vec{y}^a \mid x)\,\pi_\theta(\vec{y}^b \mid x)} \\
&\cdot \Big\{ \sigma\big(r^\star(x,\vec{y}^a) - r^\star(x,\vec{y}^b)\big) - \sigma\big(r_\theta(x,\vec{y}^a) - r_\theta(x,\vec{y}^b)\big) \Big\} \\
&\cdot \Big\{ \nabla_\theta\, r_\theta(x,\vec{y}^a) - \nabla_\theta\, r_\theta(x,\vec{y}^b) \Big\} \Bigg].
\end{aligned} \tag{20}$$

Substituting the density ratio from equation (18) into expression (20) and incorporating the weight function $w(x)$ defined in equation (10), we obtain

$$\begin{aligned}
\nabla_\theta \mathcal{L}(\theta) \;=\; &- \frac{1}{2\,\overline{Z}_\theta}\, \mathbb{E}_{x\sim\rho;\; \vec{y}^a, \vec{y}^b\sim\pi_\theta(\cdot|x)} \Bigg[ \frac{\sigma\big(r^\star(x,\vec{y}^a) - r^\star(x,\vec{y}^b)\big) - \sigma\big(r_\theta(x,\vec{y}^a) - r_\theta(x,\vec{y}^b)\big)}{\sigma'\big(r_\theta(x,\vec{y}^a) - r_\theta(x,\vec{y}^b)\big)} \\
&\cdot \Big\{ \nabla_\theta\, r_\theta(x,\vec{y}^a) - \nabla_\theta\, r_\theta(x,\vec{y}^b) \Big\} \Bigg].
\end{aligned} \tag{21}$$

Using the intuition that the first-order Taylor expansion

$$\frac{\sigma(z^\star) - \sigma(z)}{\sigma'(z)} \;=\; (z^\star - z) + \mathcal{O}\big((z^\star - z)^2\big)$$

is valid when $z \to z^\star$, with $z^\star := r^\star(x,\vec{y}^a) - r^\star(x,\vec{y}^b)$ and $z := r_\theta(x,\vec{y}^a) - r_\theta(x,\vec{y}^b)$, we find that

$$\begin{aligned}
&\frac{\sigma\big(r^\star(x,\vec{y}^a) - r^\star(x,\vec{y}^b)\big) - \sigma\big(r_\theta(x,\vec{y}^a) - r_\theta(x,\vec{y}^b)\big)}{\sigma'\big(r_\theta(x,\vec{y}^a) - r_\theta(x,\vec{y}^b)\big)} \\
&= \Big\{ \big(r^\star(x,\vec{y}^a) - r^\star(x,\vec{y}^b)\big) - \big(r_\theta(x,\vec{y}^a) - r_\theta(x,\vec{y}^b)\big) \Big\} \;+\; \text{second-order term.}
\end{aligned}$$

Reformulating equation (21) in this context, we rewrite it as

$$\begin{aligned}
\nabla_\theta \mathcal{L}(\phi) = &- \frac{1}{2\overline{Z}_\theta}\, \mathbb{E}_{\substack{x\sim\rho; \\ \vec{y}^a, \vec{y}^b\sim\pi_\theta(\cdot|x)}} \Bigg[ \Big\{ \big(r^\star(x,\vec{y}^a) - r^\star(x,\vec{y}^b)\big) - \big(r_\theta(x,\vec{y}^a) - r_\theta(x,\vec{y}^b)\big) \Big\} \\
&\cdot \Big\{ \nabla_\theta\, r_\theta(x,\vec{y}^a) - \nabla_\theta\, r_\theta(x,\vec{y}^b) \Big\} \Bigg] + T_2\,,
\end{aligned} \tag{22}$$

where $T_2$ represents the second-order residual term related to the estimation error $r_\theta - r^\star$. By applying Lemma C.2, we observe that the primary term in equation (22) aligns with the direction of $\nabla_\theta J(\pi_\theta)$, resulting in

$$\nabla_\theta \mathcal{L}(\phi) = -\frac{\beta}{\overline{Z}_\theta} \nabla_\theta J(\pi_\theta) + T_2. \tag{23}$$

Next, we proceed to control the second-order term $T_2$. The conditions

$$\|r^\star\|_\infty, \|r_\theta\|_\infty \leq R \qquad \text{and} \qquad \sup_{(x,\vec{y}) \in \mathcal{X} \times \mathcal{Y}} \|\nabla_\theta r_\theta(x, \vec{y})\|_2 \leq G,$$

lead to the bound

$$\left| \frac{\sigma(z^\star) - \sigma(z)}{\sigma'(z)} - (z^\star - z) \right| \leq 0.1 \, (1 + e^{2R}) \cdot (z^\star - z)^2,$$

which in turn implies

$$\|T_2\|_2 \leq \frac{0.1 \, (1 + e^{2R}) \, G}{\overline{Z}_\theta} \, \mathbb{E}_{x \sim \rho; \, \vec{y}^a, \vec{y}^b \sim \pi_\theta(\cdot|x)} \left[ \left\{ \left( r^\star(x, \vec{y}^a) - r^\star(x, \vec{y}^b) \right) - \left( r_\theta(x, \vec{y}^a) - r_\theta(x, \vec{y}^b) \right) \right\}^2 \right]. \tag{24}$$

Finally, combining equation (24) with equation (23), we conclude the proof of Theorem 4.1.

### C.1.3. PROOF OF CLAIM (18)

The remaining step in the proof of Theorem 4.1 is to verify the expression for the density ratio in equation (18).

Based on the sampling scheme described in Section 3, we find that the sampling distribution for the response satisfies

$$\mu(\vec{y}^a, \vec{y}^b \mid x) = \{1 - p_0(x)\} \cdot \pi_\theta(\vec{y}^a \mid x) \, \pi_\theta(\vec{y}^b \mid x) + p_0(x) \cdot \pi_\theta^+(\vec{y}^a \mid x) \, \pi_\theta^-(\vec{y}^b \mid x), \tag{25}$$

where the probability $p_0(x)$ is defined as

$$p_0(x) = Z_\theta^+(x) \, Z_\theta^-(x) / \{1 + Z_\theta^+(x) \, Z_\theta^-(x)\}$$

and the policies $\pi_\theta^+$ and $\pi_\theta^-$ are specified in equations (9a) and (9b), respectively. This allows us to simplify equation (25) to

$$\mu(\vec{y}^a, \vec{y}^b \mid x) = \frac{\pi_\theta(\vec{y}^a \mid x) \, \pi_\theta(\vec{y}^b \mid x)}{1 + Z_\theta^+(x) \, Z_\theta^-(x)} \left\{ 1 + \exp \left\{ r_\theta(x, \vec{y}^a) - r_\theta(x, \vec{y}^b) \right\} \right\}.$$

Similarly, we derive an expression for $\mu(\vec{y}^b, \vec{y}^a \mid x)$. By averaging the two expressions, for $\mu(\vec{y}^a, \vec{y}^b \mid x)$ and $\mu(\vec{y}^b, \vec{y}^a \mid x)$, we obtain

$$\frac{\overline{\mu}(\vec{y}^a, \vec{y}^b \mid x)}{\pi_\theta(\vec{y}^a \mid x) \, \pi_\theta(\vec{y}^b \mid x)} = \frac{\pi_\theta(\vec{y}^a \mid x) \, \pi_\theta(\vec{y}^b \mid x)}{2 \{1 + Z_\theta^+(x) \, Z_\theta^-(x)\}} \left\{ 2 + \exp \left\{ r_\theta(x, \vec{y}^a) - r_\theta(x, \vec{y}^b) \right\} + \exp \left\{ r_\theta(x, \vec{y}^b) - r_\theta(x, \vec{y}^a) \right\} \right\}.$$

Rewriting this expression using the formula for $\sigma'$ in equation (19), we arrive at

$$\begin{aligned}
&\{1 + Z_\theta^+(x) \, Z_\theta^-(x)\} \cdot \frac{\overline{\mu}(\vec{y}^a, \vec{y}^b \mid x)}{\pi_\theta(\vec{y}^a \mid x) \, \pi_\theta(\vec{y}^b \mid x)} \\
&= \frac{1}{2} \left\{ 1 + \exp \left\{ r_\theta(x, \vec{y}^b) - r_\theta(x, \vec{y}^a) \right\} \right\} \left\{ 1 + \exp \left\{ r_\theta(x, \vec{y}^a) - r_\theta(x, \vec{y}^b) \right\} \right\} \\
&= \frac{1}{2 \, \sigma' \left( r_\theta(x, \vec{y}^a) - r_\theta(x, \vec{y}^b) \right)}.
\end{aligned}$$

Finally, rearranging terms, we recover equation (18), completing this part of the proof.

## C.2. Statistical Considerations

In this section, we present the proofs for Theorems 4.2 and 4.3 and Lemma B.1 from Section 4.2. We start with the proof of Lemma B.1 in Appendix C.2.1, with a rigorous restatement provided in Theorem C.4 below.

**Theorem C.4.** *Assume the reward model $r^\star$ in the BT model (1) satisfies $r^\star = r_{\theta^\star}$ for some parameter $\theta^\star$. Assume that $\widehat{\theta}$ minimizes the loss function $\widehat{\mathcal{L}}(\theta)$ in the sense that $\sqrt{n}\,\nabla_\theta\,\widehat{\mathcal{L}}(\widehat{\theta}) \xrightarrow{p} \mathbf{0}$ and that $\widehat{\theta} \xrightarrow{p} \theta^\star$ as the sample size $n \to \infty$. Additionally, suppose the reward function $r_\theta(x, \vec{y})$, its gradient $\nabla_\theta\, r_\theta(x, \vec{y})$ and its Hessian $\nabla_\theta^2\, r_\theta(x, \vec{y})$ are uniformly bounded and Lipchitz continuous with respect to $\theta$, for all $(x, \vec{y}) \in \mathcal{X} \times \mathcal{Y}$.*

*Under these conditions, the estimate $\widehat{\theta}$ asymptotically follows a Gaussian distribution*

$$\sqrt{n}\,(\widehat{\theta} - \theta^\star) \xrightarrow{d} \mathcal{N}(\mathbf{0}, \mathbf{\Omega}) \qquad \text{as } n \to \infty\,.$$

*We have an estimate of the covariance matrix $\mathbf{\Omega}$:*

$$\mathbf{\Omega} \preceq \|w\|_\infty \cdot \mathbf{\Sigma}_\star^{-1}\,.$$

*For a general sampling scheme $\mu$ chosen, the matrix $\mathbf{\Sigma}_\star$ is given by*

$$\mathbf{\Sigma}_\star := \mathbb{E}_{x \sim \rho,\, (\vec{y}^a, \vec{y}^b) \sim \overline{\mu}(\cdot | x)} \Big[ w(x) \cdot \mathrm{Var}\big(\mathbb{1}\{\vec{y}^a = \vec{y}^w\} \mid x, \vec{y}^a, \vec{y}^b\big) \cdot \boldsymbol{g}\,\boldsymbol{g}^\top \Big]\,,$$

*where the expectation is taken over the distribution*

$$\overline{\mu}(\vec{y}^a, \vec{y}^b \mid x) := \frac{1}{2}\left\{ \mu(\vec{y}^a, \vec{y}^b \mid x) + \mu(\vec{y}^b, \vec{y}^a \mid x) \right\}\,.$$

*The variance term is specified as*

$$\mathrm{Var}\big(\mathbb{1}\{\vec{y}^a = \vec{y}^w\} \mid x, \vec{y}^a, \vec{y}^b\big) = \sigma\big(r^\star(x, \vec{y}^a) - r^\star(x, \vec{y}^b)\big)\,\sigma\big(r^\star(x, \vec{y}^b) - r^\star(x, \vec{y}^a)\big)$$

*and the gradient difference $\boldsymbol{g}$ is defined as*

$$\boldsymbol{g} := \nabla_\theta\, r^\star(x, \vec{y}^a) - \nabla_\theta\, r^\star(x, \vec{y}^b)\,.$$

Theorem C.4 establishes the asymptotic distribution of the estimated parameter $\widehat{\theta}$, which serves as the foundation for the subsequent results. Next, we show that Theorem 4.2 directly follows as a corollary of Theorem C.4, with the detailed derivation provided in Appendix C.2.2. Finally, in Appendix C.2.3, we prove Theorem 4.3, which describes the asymptotic behavior of the value gap $J(\pi^\star) - J(\widehat{\pi})$.

### C.2.1. PROOF OF LEMMA B.1 (THEOREM C.4)

In this section, we analyze the asymptotic distribution of the estimated parameter $\widehat{\theta}$ for a general sampling distribution $\mu$. The parameter $\widehat{\theta}$ is obtained by solving the optimization problem

$$\text{minimize}_\theta \quad \widehat{\mathcal{L}}(\theta) := -\frac{1}{n}\sum_{i=1}^n w(x_i) \cdot \log \sigma\Big(r_\theta\big(x_i, \vec{y}_i^w\big) - r_\theta\big(x_i, \vec{y}_i^\ell\big)\Big)\,.$$

We assume the optimization is performed to sufficient accuracy such that $\nabla_\theta\,\widehat{\mathcal{L}}(\widehat{\theta}) = o_p\big(n^{-\frac{1}{2}}\big)$. Under this condition, $\widehat{\theta}$ qualifies as a $Z$-estimator. To study its asymptotic behavior, we use the master theorem for $Z$-estimators (Kosorok, 2008), the formal statement of which is provided in Theorem E.1 in Appendix E.

To apply the master theorem, we set $\Psi := \nabla_\theta\,\mathcal{L}$ and $\Psi_n := \nabla_\theta\,\widehat{\mathcal{L}}$ and verify the conditions. In particular, the smoothness condition (64) in Theorem E.1 translates to the following equation in our context:

$$\sqrt{n}\left\{\nabla_\theta\,\widehat{\mathcal{L}}(\widehat{\theta}) - \nabla_\theta\,\mathcal{L}(\widehat{\theta})\right\} - \sqrt{n}\left\{\nabla_\theta\,\widehat{\mathcal{L}}(\theta^\star) - \nabla_\theta\,\mathcal{L}(\theta^\star)\right\} = o_p\big(1 + \sqrt{n}\,\|\widehat{\theta} - \theta^\star\|_2\big)\,. \tag{27}$$

This condition follows from the second-order smoothness of the reward function $r_\theta$ with respect to $\theta$. A rigorous proof is provided in Appendix D.2.1.

We now provide the explicit form of the derivative $\dot{\Psi}_{\theta^\star} = \nabla_\theta^2\,\mathcal{L}(\theta^\star)$, as captured in the following lemma. The proof of this result can be found in Appendix D.2.2.

**Lemma C.5.** *The Hessian matrix of the population loss $\mathcal{L}(\theta)$ at $\theta = \theta^\star$ is*

$$\nabla^2_\theta \mathcal{L}(\theta^\star) \;=\; \Sigma_\star, \tag{28}$$

*where the matrix $\Sigma_\star$ is defined in equation (14).*

Next, we analyze the asymptotic behavior of the gradient $\nabla_\theta \widehat{\mathcal{L}}(\theta^\star)$. The proof is deferred to Appendix D.2.3.

**Lemma C.6.** *The gradient of the empirical loss $\widehat{\mathcal{L}}(\theta)$ at $\theta = \theta^\star$ satisfies*

$$\sqrt{n}\left(\nabla_\theta \widehat{\mathcal{L}}(\theta^\star) - \nabla_\theta \mathcal{L}(\theta^\star)\right) \;\xrightarrow{d}\; \mathcal{N}(\mathbf{0}, \widetilde{\Omega}) \qquad as\ n \to \infty, \tag{29a}$$

*where the covariance matrix $\widetilde{\Omega} \in \mathbb{R}^{d \times d}$ is bounded as follows:*

$$\widetilde{\Omega} \;\preceq\; \|w\|_\infty \cdot \Sigma_\star, \tag{29b}$$

*with $\Sigma_\star$ defined in equation (14).*

Combining these results, and assuming $\Sigma_\star$ is nonsingular, the master theorem (Theorem E.1) yields the asymptotic distribution of $\widehat{\theta}$:

$$\sqrt{n}\left(\widehat{\theta} - \theta^\star\right) \;\xrightarrow{d}\; \mathcal{N}\left(\mathbf{0}, \Sigma_\star^{-1}\widetilde{\Omega}\Sigma_\star^{-1}\right).$$

Furthermore, from the bound (29b), the covariance matrix $\Omega;:= \Sigma_\star^{-1}\widetilde{\Omega}\Sigma_\star^{-1}$ satisfies

$$\Omega \;=\; \Sigma_\star^{-1}\widetilde{\Omega}\Sigma_\star^{-1} \;\preceq\; \|w\|_\infty \cdot \Sigma_\star^{-1}.$$

Therefore, we have established the asymptotic distribution of $\widehat{\theta}$, completing the proof of Lemma B.1.

### C.2.2. PROOF OF THEOREM 4.2

Theorem 4.2 is a direct corollary of Lemma B.1, using our specific choice of sampling distribution $\mu$. To establish this, we demonstrate how the general covariance matrix $\Sigma_\star$ in equation (14) simplifies to the form in equation (11) under our proposed sampling scheme.

To establish the result in this section, we impose the following regularity condition: There exists a constant $C \geq 1$ satisfying

$$\text{Var}_{r_\theta}\left(\mathbb{1}\{\vec{y}^a = \vec{y}^w\} \mid x, \vec{y}^a, \vec{y}^b\right) \;\leq\; C \cdot \text{Var}_{r^\star}\left(\mathbb{1}\{\vec{y}^a = \vec{y}^w\} \mid x, \vec{y}^a, \vec{y}^b\right) \tag{30}$$

for any prompt $x \in \mathcal{X}$ and responses $\vec{y}^a, \vec{y}^b \in \mathcal{Y}$. Here $\text{Var}_{r_\theta}\left(\mathbb{1}\{\vec{y}^a = \vec{y}^w\} \mid x, \vec{y}^a, \vec{y}^b\right)$ denotes the conditional variance under the BT model (1), when the implicit reward function $r^\star$ is replaced by $r_\theta$. The term $\text{Var}_{r^\star}\left(\mathbb{1}\{\vec{y}^a = \vec{y}^w\} \mid x, \vec{y}^a, \vec{y}^b\right) \equiv \text{Var}\left(\mathbb{1}\{\vec{y}^a = \vec{y}^w\} \mid x, \vec{y}^a, \vec{y}^b\right)$ represents the conditional variance under the ground-truth BT model, where the reward function is given by $r^\star$.

We begin by leveraging the property of the sampling distribution $\mu$ from equation (18) and the derivative $\sigma'$ of the sigmoid function $\sigma$, given in equation (19). Specifically, we find that

$$
\begin{aligned}
\frac{\overline{\mu}(\vec{y}^a, \vec{y}^b \mid x)}{\pi_\theta(\vec{y}^a \mid x)\,\pi_\theta(\vec{y}^b \mid x)} &= \frac{1}{2\left\{1 + Z_\theta^+(x)\,Z_\theta^-(x)\right\}} \cdot \frac{1}{\sigma\left(r_\theta(x, \vec{y}^a) - r_\theta(x, \vec{y}^b)\right)\,\sigma\left(r_\theta(x, \vec{y}^b) - r_\theta(x, \vec{y}^a)\right)} \\
&= \frac{1}{2\left\{1 + Z_\theta^+(x)\,Z_\theta^-(x)\right\}} \cdot \frac{1}{\text{Var}_{r_\theta}\left(\mathbb{1}\{\vec{y}^a = \vec{y}^w\} \mid x, \vec{y}^a, \vec{y}^b\right)}.
\end{aligned}
$$

We then apply condition (30) and derive

$$\frac{\overline{\mu}(\vec{y}^a, \vec{y}^b \mid x)}{\pi_\theta(\vec{y}^a \mid x)\,\pi_\theta(\vec{y}^b \mid x)} \;\geq\; \frac{C^{-1}}{2\left\{1 + Z_\theta^+(x)\,Z_\theta^-(x)\right\}} \cdot \frac{1}{\text{Var}_{r^\star}\left(\mathbb{1}\{\vec{y}^a = \vec{y}^w\} \mid x, \vec{y}^a, \vec{y}^b\right)}. \tag{31}$$

Next, substituting this result (31) into equation (14), alongside the weight function $w(x)$ from equation (10), we reform $\boldsymbol{\Sigma}_\star$ as

$$
\begin{aligned}
\boldsymbol{\Sigma}_\star &= \mathbb{E}_{x \sim \rho; \, \vec{\boldsymbol{y}}^a, \vec{\boldsymbol{y}}^b \sim \pi_\theta(\cdot | x)} \left[ \frac{\overline{\mu}(\vec{\boldsymbol{y}}^a, \vec{\boldsymbol{y}}^b \mid x)}{\pi_\theta(\vec{\boldsymbol{y}}^a \mid x) \, \pi_\theta(\vec{\boldsymbol{y}}^b \mid x)} \cdot w(x) \cdot \mathrm{Var}\big(\mathbb{1}\{\vec{\boldsymbol{y}}^a = \vec{\boldsymbol{y}}^w\} \mid x, \vec{\boldsymbol{y}}^a, \vec{\boldsymbol{y}}^b\big) \cdot \boldsymbol{g} \, \boldsymbol{g}^\top \right] \\
&\succeq \frac{1}{2 \, C \, \overline{Z}_\theta} \, \mathbb{E}_{x \sim \rho; \, \vec{\boldsymbol{y}}^a, \vec{\boldsymbol{y}}^b \sim \pi_\theta(\cdot | x)} \big[ \boldsymbol{g} \, \boldsymbol{g}^\top \big].
\end{aligned}
\tag{32}
$$

The conditional expectation of $\boldsymbol{g}\boldsymbol{g}^\top$ simplifies as

$$
\begin{aligned}
&\mathbb{E}_{\vec{\boldsymbol{y}}^a, \vec{\boldsymbol{y}}^b \sim \pi_\theta(\cdot|x)} \big[ \boldsymbol{g}\boldsymbol{g}^\top \mid x \big] \\
&= \mathbb{E}_{\vec{\boldsymbol{y}}^a, \vec{\boldsymbol{y}}^b \sim \pi_\theta(\cdot|x)} \Big[ \big\{ \nabla_\theta \, r^\star(x, \vec{\boldsymbol{y}}^a) - \nabla_\theta \, r^\star(x, \vec{\boldsymbol{y}}^b) \big\} \big\{ \nabla_\theta \, r^\star(x, \vec{\boldsymbol{y}}^a) - \nabla_\theta \, r^\star(x, \vec{\boldsymbol{y}}^b) \big\}^\top \mid x \Big] \\
&= 2 \cdot \mathbb{E}_{\vec{\boldsymbol{y}} \sim \pi_\theta(\cdot|x)} \Big[ \nabla_\theta \, r^\star(x, \vec{\boldsymbol{y}}) \, \nabla_\theta \, r^\star(x, \vec{\boldsymbol{y}})^\top \mid x \Big] - 2 \cdot \mathbb{E}_{\vec{\boldsymbol{y}} \sim \pi_\theta(\cdot|x)} \big[ \nabla_\theta \, r^\star(x, \vec{\boldsymbol{y}}) \mid x \big] \, \mathbb{E}_{\vec{\boldsymbol{y}} \sim \pi_\theta(\cdot|x)} \big[ \nabla_\theta \, r^\star(x, \vec{\boldsymbol{y}}) \mid x \big]^\top \\
&= 2 \cdot \mathrm{Cov}_{\vec{\boldsymbol{y}} \sim \pi_\theta(\cdot|x)} \big[ \nabla_\theta \, r^\star(x, \vec{\boldsymbol{y}}) \mid x \big].
\end{aligned}
$$

Substituting this result into equation (32), we arrive at the conclusion that

$$
\boldsymbol{\Sigma}_\star \succeq \frac{1}{C \, \overline{Z}_\phi} \, \mathbb{E}_{x \sim \rho} \Big[ \mathrm{Cov}_{\vec{\boldsymbol{y}} \sim \pi^\star(\cdot|x)} \big[ \nabla_\theta \, r^\star(x, \vec{\boldsymbol{y}}) \mid x \big] \Big],
$$

which matches the simplified form in equation (11) as stated in Theorem 4.2.

### C.2.3. PROOF OF THEOREM 4.3

**Gradient $\nabla_\theta J(\pi^\star)$ and Hessian $\nabla_\theta^2 J(\pi^\star)$:** The equality $\nabla_\theta J(\pi^\star) = 0$ follows directly from the gradient expression (41) for $\nabla_\theta J(\pi_\theta)$, evaluated at $\theta = \theta^\star$ with $r_\theta = r^\star$.

The proof of the Hessian result, $\nabla_\theta^2 J(\pi^\star) = -(1/\beta) \cdot \boldsymbol{\Sigma}_\star$, involves straightforward but technical differentiation of equation (41). For brevity, we defer this proof to Appendix D.3.1.

**Asymptotic Distribution of Value Gap $J(\pi^\star) - J(\widehat{\pi})$:** To understand the behavior of the value gap $J(\pi^\star) - J(\widehat{\pi})$, we start by applying a Taylor expansion of $J(\pi_\theta)$ around $\theta^\star$. This gives

$$
J(\pi^\star) - J(\widehat{\pi}) = \nabla_\theta J(\pi^\star)^\top (\theta^\star - \widehat{\theta}) - \frac{1}{2} (\theta^\star - \widehat{\theta})^\top \nabla_\theta^2 J(\pi^\star) (\theta^\star - \widehat{\theta}) + o\big(\|\theta^\star - \widehat{\theta}\|_2^2\big).
$$

By substituting $\nabla_\theta J(\pi^\star) = \boldsymbol{0}$ (a direct result of the optimality of $\pi^\star$), the linear term vanishes. Introducing the shorthand $\boldsymbol{H} := -\nabla_\theta^2 J(\pi^\star) = (1/\beta) \cdot \boldsymbol{\Sigma}_\star$, the expression simplifies to

$$
J(\pi^\star) - J(\widehat{\pi}) = \frac{1}{2} (\widehat{\theta} - \theta^\star)^\top \boldsymbol{H} (\widehat{\theta} - \theta^\star) + o\big(\|\widehat{\theta} - \theta^\star\|_2^2\big).
\tag{33}
$$

When the sample size $n$ is sufficiently large, $\widehat{\theta}$ approaches $\theta^\star$, making the higher-order term negligible. Therefore, the value gap is dominated by the quadratic form.

From Theorem 4.2, we know the parameter estimate $\widehat{\theta}$ satisfies

$$
\sqrt{n} \, (\widehat{\theta} - \theta^\star) \xrightarrow{d} \mathcal{N}(\boldsymbol{0}, \boldsymbol{\Omega}).
$$

Substituting this result into the quadratic approximation of the value gap, we find that the scaled value gap has the asymptotic distribution

$$
n \cdot \{ J(\pi^\star) - J(\widehat{\pi}) \} \xrightarrow{d} \frac{1}{2} \boldsymbol{z}^\top \boldsymbol{\Omega}^{\frac{1}{2}} \boldsymbol{H} \boldsymbol{\Omega}^{\frac{1}{2}} \boldsymbol{z} =: \boldsymbol{X} \qquad \text{where } \boldsymbol{z} \sim \mathcal{N}(\boldsymbol{0}, \boldsymbol{I}).
\tag{34}
$$

This approximation provides a clear intuition: the value gap is asymptotically driven by a weighted chi-squared-like term involving the covariance structure $\boldsymbol{\Omega}$ and the Hessian-like matrix $\boldsymbol{H}$.

To rigorously establish this result, we will apply Slutsky's theorem. The full proof is presented in Appendix D.3.2.

**Bounding the Chi-Square Distribution:** To bound the random variable $X$, we first leverage the estimate of the covariance matrix $\Omega$ provided by Theorem 4.2:

$$\Omega \preceq C \overline{Z}_\theta \|w\|_\infty \cdot \Sigma_\star^{-1},$$

where the constant $C$ comes from condition (30). It follows that the matrix $\Omega^{\frac{1}{2}} H \Omega^{\frac{1}{2}}$ appearing in equation (34) can be bounded as

$$\Omega^{\frac{1}{2}} H \Omega^{\frac{1}{2}} \preceq C \|w\|_\infty \cdot \Sigma_\star^{-\frac{1}{2}} H \Sigma_\star^{-\frac{1}{2}} = C \cdot \frac{\overline{Z}_\theta \|w\|_\infty}{\beta} \cdot I = C \cdot \frac{1 + \|Z_\theta^+ Z_\theta^-\|_\infty}{\beta} \cdot I.$$

Here the last equality uses the definition of the weight function $w$ from equation (10). Substituting this bound into the quadratic form, we derive

$$X = \frac{1}{2} z^\top \Omega^{\frac{1}{2}} H \Omega^{\frac{1}{2}} z \leq C \cdot \frac{1 + \|Z_\theta^+ Z_\theta^-\|_\infty}{2\beta} \cdot z^\top z,$$

where $z \sim \mathcal{N}(\mathbf{0}, I)$. Since $z^\top z$ follows a chi-square distribution with $d$ degrees of freedom, $X$ is stochastically dominated by a rescaled chi-square random variable

$$C \cdot \frac{1 + \|Z_\theta^+ Z_\theta^-\|_\infty}{2\beta} \cdot \chi_d^2.$$

Equivalently, we can express this dominance as

$$\limsup_{n \to \infty} \mathbb{P}\left\{ n \left\{ J(\pi^\star) - J(\widehat{\pi}) \right\} > C \cdot \frac{1 + \|Z_\theta^+ Z_\theta^-\|_\infty}{2\beta} \cdot t \right\} \leq \mathbb{P}\{\chi_d^2 > t\} \qquad \text{for any } t > 0. \tag{35}$$

This inequality, given in equation (35), corresponds to the first bound in equation (13).

The second inequality in equation (13) provides a precise tail bound for $\chi_d^2$. As its proof involves more technical details, we defer it to Appendix D.3.3.

# D. Proof of Auxiliary Results

This section provides proofs of auxiliary results supporting the main theorems and lemmas. In Appendix D.1, we present the auxiliary results required for Theorem 4.1. Appendix D.2 details the proofs of supporting results for Theorem 4.2. Finally, in Appendix D.3, we establish the auxiliary results necessary for Theorem 4.3.

## D.1. Proof of Auxiliary Results for Theorem 4.1

In this section, we provide the proofs of several auxiliary results that support the proof of Theorem 4.1. Specifically, Appendix D.1.1 presents the forms of the gradients of the policy $\pi_\theta$ and the reward $r_\theta$, which serve as fundamental building blocks for deriving the lemmas. Appendix D.1.2 analyzes the gradient of the return function $J(\pi_\theta)$, as defined in equation (6). Appendix D.1.3 focuses on deriving expressions for the gradient of the negative log-likelihood function $\mathcal{L}(\theta)$.

### D.1.1. GRADIENTS OF POLICY $\pi_\theta$ AND REWARD $r_\theta$

In this part, we introduce results for the gradients of policy $\pi_\theta$ and reward $r_\theta$ with respsect to parameter $\theta$, which lay the foundation of our calculations.

**Lemma D.1** (Gradients of policy $\pi_\theta$ and reward function $r_\theta$). *The gradients of the policy $\pi_\theta$ and the reward function $r_\theta$ can be expressed in terms of each other as follows*

$$\nabla_\theta \pi_\theta(d\vec{y} \mid x) = \pi_\theta(d\vec{y} \mid x) \cdot \frac{1}{\beta} \left\{ \nabla_\theta r_\theta(x, \vec{y}) - \mathbb{E}_{\vec{y}' \sim \pi_\theta(\cdot \mid x)} \left[ \nabla_\theta r_\theta(x, \vec{y}') \right] \right\}, \tag{36a}$$

$$\nabla_\theta r_\theta(x, \vec{y}) = \beta \cdot \frac{\nabla_\theta \pi_\theta(\vec{y} \mid x)}{\pi_\theta(\vec{y} \mid x)}. \tag{36b}$$

We now proceed to prove Lemma D.1.

To begin, recall our definition of the reward function $r_\theta$ as given in equation (5). It directly follows that

$$\nabla_\theta \, r_\theta(x, \vec{y}) \;=\; \beta \cdot \frac{\nabla_\theta \, \pi_\theta(\vec{y} \mid x)}{\pi_\theta(\vec{y} \mid x)} \,.$$

This result confirms equation (36b) as stated in Lemma D.1.

Next, we express the policy $\pi_\theta(d\vec{y} \mid x)$ in terms of the reward function $r_\theta(x, \vec{y})$. By reformulating equation (5), we obtain

$$\pi_\theta(d\vec{y} \mid x) \;=\; \frac{1}{Z_\theta(x)} \, \pi_{\text{ref}}(d\vec{y} \mid x) \exp\left\{ \frac{1}{\beta} \, r_\theta(x, \vec{y}) \right\}, \tag{37a}$$

where $Z_\theta(x)$ is the partition function defined as

$$Z_\theta(x) \;=\; \int_{\mathcal{Y}} \pi_{\text{ref}}(d\vec{y} \mid x) \exp\left\{ \frac{1}{\beta} \, r_\theta(x, \vec{y}) \right\}. \tag{37b}$$

We then compute the gradient of $\pi_\theta(d\vec{y} \mid x)$ with respect to $\theta$. Applying the chain rule, we get

$$\begin{aligned}
\nabla_\theta \, \pi_\theta(d\vec{y} \mid x) \;=\;\; & \frac{1}{Z_\theta(x)} \, \pi_{\text{ref}}(d\vec{y} \mid x) \exp\left\{ \frac{1}{\beta} \, r_\theta(x, \vec{y}) \right\} \cdot \frac{1}{\beta} \, \nabla_\theta \, r_\theta(x, \vec{y}) \\
& - \frac{1}{Z_\theta^2(x)} \, \pi_{\text{ref}}(d\vec{y} \mid x) \exp\left\{ \frac{1}{\beta} \, r_\theta(x, \vec{y}) \right\} \cdot \nabla_\theta \, Z_\theta(x) \,.
\end{aligned} \tag{38}$$

We need the gradient of the partition function $Z_\theta(x)$:

$$\begin{aligned}
\nabla_\theta \, Z_\theta(x) \;&=\; \int_{\mathcal{Y}} \pi_{\text{ref}}(d\vec{y} \mid x) \exp\left\{ \frac{1}{\beta} \, r_\theta(x, \vec{y}) \right\} \cdot \frac{1}{\beta} \, \nabla_\theta \, r_\theta(x, \vec{y}) \\
&=\; Z_\theta(x) \cdot \int_{\mathcal{Y}} \pi_\theta(d\vec{y} \mid x) \cdot \frac{1}{\beta} \, \nabla_\theta \, r_\theta(x, \vec{y}) \\
&=\; Z_\theta(x) \cdot \frac{1}{\beta} \, \mathbb{E}_{\vec{y} \sim \pi_\theta(\cdot \mid x)} \big[ \nabla_\theta \, r_\theta(x, \vec{y}) \big] \,.
\end{aligned} \tag{39}$$

Substituting equation (39) back into equation (38), we simplify the expression for the gradient of $\pi_\theta(d\vec{y} \mid x)$:

$$\nabla_\theta \, \pi_\theta(d\vec{y} \mid x) \;=\; \frac{1}{Z_\theta(x)} \, \pi_{\text{ref}}(d\vec{y} \mid x) \exp\left\{ \frac{1}{\beta} \, r_\theta(x, \vec{y}) \right\} \cdot \frac{1}{\beta} \left\{ \nabla_\theta \, r_\theta(x, \vec{y}) - \mathbb{E}_{\vec{y}' \sim \pi_\theta(\cdot \mid x)} \big[ \nabla_\theta \, r_\theta(x, \vec{y}') \big] \right\}.$$

This matches equation (36a) from Lemma D.1, thereby completing the proof.

### D.1.2. PROOF OF LEMMA C.2

Equality (16) in Lemma C.2 can be derived as a consequence of a more detailed result. We state it in Lemma D.2.

**Lemma D.2.** *For a policy $\pi_\theta$, the gradients with respect to the parameter $\theta$ of its expected return $\mathbb{E}_{x \sim \rho, \, \vec{y} \sim \pi_\theta(\cdot \mid x)} \big[ r^\star(x, \vec{y}) \big]$ and its KL divergence from a reference policy $D_{KL}(\pi_\theta \parallel \pi_{\text{ref}})$ are given by*

$$\nabla_\theta \, \mathbb{E}_{x \sim \rho, \, \vec{y} \sim \pi_\theta(\cdot \mid x)} \big[ r^\star(x, \vec{y}) \big] = \frac{1}{\beta} \, \mathbb{E}_{x \sim \rho, \, \vec{y} \sim \pi_\theta(\cdot \mid x)} \left[ r^\star(x, \vec{y}) \Big\{ \nabla_\theta \, r_\theta(x, \vec{y}) - \mathbb{E}_{\vec{y}' \sim \pi_\theta(\cdot \mid x)} \big[ \nabla_\theta \, r_\theta(x, \vec{y}') \big] \Big\} \right], \tag{40a}$$

$$\nabla_\theta \, D_{KL}(\pi_\theta \parallel \pi_{\text{ref}}) = \frac{1}{\beta^2} \, \mathbb{E}_{x \sim \rho, \, \vec{y} \sim \pi_\theta(\cdot \mid x)} \left[ r_\theta(x, \vec{y}) \Big\{ \nabla_\theta \, r_\theta(x, \vec{y}) - \mathbb{E}_{\vec{y}' \sim \pi_\theta(\cdot \mid x)} \big[ \nabla_\theta \, r_\theta(x, \vec{y}') \big] \Big\} \right]. \tag{40b}$$

Recall that the scalar value $J(\pi_\theta)$ of the policy is defined as

$$J(\pi_\theta) \;=\; \mathbb{E}_{x \sim \rho, \, \vec{y} \sim \pi_\theta(\cdot \mid x)} \big[ r^\star(x, \vec{y}) \big] \;-\; \beta \, D_{\text{KL}}(\pi_\theta \parallel \pi_{\text{ref}}) \,.$$

Using Lemma D.2, we derive the gradient of $J(\pi_\theta)$ as

$$
\begin{aligned}
\nabla_\theta J(\pi_\theta) &= \nabla_\theta \, \mathbb{E}_{x \sim \rho, \, \vec{y} \sim \pi_\theta(\cdot|x)}\big[r^\star(x, \vec{y})\big] - \beta \, \nabla_\theta \, D_{\mathrm{KL}}(\pi_\theta \parallel \pi_{\mathrm{ref}}) \\
&= \frac{1}{\beta} \, \mathbb{E}_{x \sim \rho, \, \vec{y} \sim \pi_\theta(\cdot|x)}\Big[\big\{r^\star(x, \vec{y}) - r_\theta(x, \vec{y})\big\}\big\{\nabla_\theta \, r_\theta(x, \vec{y}) - \mathbb{E}_{\vec{y}' \sim \pi_\theta(\cdot|x)}\big[\nabla_\theta \, r_\theta(x, \vec{y}')\big]\big\}\Big].
\end{aligned}
\tag{41}
$$

We rewrite the expression in equation (41) in two equivalent forms by exchanging the roles of $\vec{y}^a$ and $\vec{y}^b$:

$$
\nabla_\theta J(\pi_\theta) = \frac{1}{\beta} \, \mathbb{E}_{x \sim \rho, \, \vec{y}^a \sim \pi_\theta(\cdot|x)}\Big[\big\{r^\star(x, \vec{y}^a) - r_\theta(x, \vec{y}^a)\big\}\big\{\nabla_\theta \, r_\theta(x, \vec{y}^a) - \mathbb{E}_{\vec{y}^b \sim \pi_\theta(\cdot|x)}\big[\nabla_\theta \, r_\theta(x, \vec{y}^b)\big]\big\}\Big],
\tag{42a}
$$

$$
\nabla_\theta J(\pi_\theta) = \frac{1}{\beta} \, \mathbb{E}_{x \sim \rho, \, \vec{y}^b \sim \pi_\theta(\cdot|x)}\Big[\big\{r^\star(x, \vec{y}^b) - r_\theta(x, \vec{y}^b)\big\}\big\{\nabla_\theta \, r_\theta(x, \vec{y}^b) - \mathbb{E}_{\vec{y}^a \sim \pi_\theta(\cdot|x)}\big[\nabla_\theta \, r_\theta(x, \vec{y}^a)\big]\big\}\Big].
\tag{42b}
$$

By taking the average of the two equivalent formulations above, we obtain equality (16) and complete the proof of Lemma C.2.

We now proceed to prove Lemma D.2, tackling equalities (40a) and (40b) one by one.

**Proof of Equality (40a) from Lemma D.2:** We begin by expressing the expected return as

$$
\mathbb{E}_{x \sim \rho, \, \vec{y} \sim \pi_\theta(\cdot|x)}\big[r^\star(x, \vec{y})\big] = \mathbb{E}_{x \sim \rho}\bigg[\int_{\mathcal{Y}} r^\star(x, \vec{y}) \, \pi_\theta(d\vec{y} \mid x)\bigg].
$$

Taking the gradient of both sides with respect to $\theta$, we have

$$
\nabla_\theta \, \mathbb{E}_{x \sim \rho, \, \vec{y} \sim \pi_\theta(\cdot|x)}\big[r^\star(x, \vec{y})\big] = \mathbb{E}_{x \sim \rho}\bigg[\int_{\mathcal{Y}} r^\star(x, \vec{y}) \, \nabla_\theta \, \pi_\theta(d\vec{y} \mid x)\bigg].
\tag{43}
$$

Using the expression for the policy gradient $\nabla_\theta \, \pi_\theta$ provided in Lemma D.1, the right-hand side of (43) simplifies to

$$
\begin{aligned}
\text{RHS of (43)} &= \mathbb{E}_{x \sim \rho}\bigg[\int_{\mathcal{Y}} r^\star(x, \vec{y}) \, \pi_\theta(d\vec{y} \mid x) \cdot \frac{1}{\beta}\Big\{\nabla_\theta \, r_\theta(x, \vec{y}) - \mathbb{E}_{\vec{y}' \sim \pi_\theta(\cdot|x)}\big[\nabla_\theta \, r_\theta(x, \vec{y}')\big]\Big\}\bigg] \\
&= \frac{1}{\beta} \, \mathbb{E}_{x \sim \rho, \, \vec{y} \sim \pi_\theta(\cdot|x)}\bigg[r^\star(x, \vec{y})\Big\{\nabla_\theta \, r_\theta(x, \vec{y}) - \mathbb{E}_{\vec{y}' \sim \pi_\theta(\cdot|x)}\big[\nabla_\theta \, r_\theta(x, \vec{y}')\big]\Big\}\bigg].
\end{aligned}
$$

This completes the verification of equation (40a) from Lemma C.2.

**Proof of Equality (40b) from Lemma D.2:** Recall the definition of the KL divergence

$$
D_{\mathrm{KL}}(\pi_\theta \parallel \pi_{\mathrm{ref}}) = \mathbb{E}_{x \sim \rho}\bigg[\int_{\mathcal{Y}} \pi_\theta(d\vec{y} \mid x) \log\bigg(\frac{\pi_\theta(\vec{y} \mid x)}{\pi_{\mathrm{ref}}(\vec{y} \mid x)}\bigg)\bigg].
$$

Applying the chain rule, we obtain

$$
\nabla_\theta \, D_{\mathrm{KL}}(\pi_\theta \parallel \pi_{\mathrm{ref}}) = \mathbb{E}_{x \sim \rho}\bigg[\int_{\mathcal{Y}} \nabla_\theta \, \pi_\theta(d\vec{y} \mid x) \log\bigg(\frac{\pi_\theta(\vec{y} \mid x)}{\pi_{\mathrm{ref}}(\vec{y} \mid x)}\bigg)\bigg] + \mathbb{E}_{x \sim \rho}\bigg[\int_{\mathcal{Y}} \nabla_\theta \, \pi_\theta(d\vec{y} \mid x)\bigg].
\tag{44}
$$

Since the policy integrates to 1, i.e., $\int_{\mathcal{Y}} \pi_\theta(d\vec{y} \mid x) = 1$, it always holds that

$$
\int_{\mathcal{Y}} \nabla_\theta \, \pi_\theta(d\vec{y} \mid x) = \nabla_\theta \int_{\mathcal{Y}} \pi_\theta(d\vec{y} \mid x) = 0,
\tag{45}
$$

i.e., the second term on the right-hand side of (44) is zero. Using the expression (37a), we take the logarithm

$$
\log\bigg(\frac{\pi_\theta(\vec{y} \mid x)}{\pi_{\mathrm{ref}}(\vec{y} \mid x)}\bigg) = \frac{1}{\beta} \, r_\theta(x, \vec{y}) - \log Z_\theta(x).
\tag{46}
$$

Combining equations (45) and (46), we get

$$\int_{\mathcal{Y}} \nabla_\theta \pi_\theta(d\vec{\boldsymbol{y}} \mid x) \log\left(\frac{\pi_\theta(\vec{\boldsymbol{y}} \mid x)}{\pi_{\text{ref}}(\vec{\boldsymbol{y}} \mid x)}\right)$$

$$= \frac{1}{\beta} \int_{\mathcal{Y}} r_\theta(x, \vec{\boldsymbol{y}}) \nabla_\theta \pi_\theta(d\vec{\boldsymbol{y}} \mid x) - \log Z_\theta(x) \int_{\mathcal{Y}} \nabla_\theta \pi_\theta(d\vec{\boldsymbol{y}} \mid x)$$

$$= \frac{1}{\beta} \int_{\mathcal{Y}} r_\theta(x, \vec{\boldsymbol{y}}) \nabla_\theta \pi_\theta(d\vec{\boldsymbol{y}} \mid x). \tag{47}$$

Now, similar to the proof of equation (40a), we derive

$$\text{RHS of (44)} = \frac{1}{\beta} \mathbb{E}_{x \sim \rho}\left[\int_{\mathcal{Y}} r_\theta(x, \vec{\boldsymbol{y}}) \nabla_\theta \pi_\theta(d\vec{\boldsymbol{y}} \mid x)\right]$$

$$= \frac{1}{\beta^2} \mathbb{E}_{x \sim \rho,\, \vec{\boldsymbol{y}} \sim \pi_\theta(\cdot \mid x)}\left[r_\theta(x, \vec{\boldsymbol{y}})\left\{\nabla_\theta r_\theta(x, \vec{\boldsymbol{y}}) - \mathbb{E}_{\vec{\boldsymbol{y}}' \sim \pi_\theta(\cdot \mid x)}\left[\nabla_\theta r_\theta(x, \vec{\boldsymbol{y}}')\right]\right\}\right],$$

which verifies equality (40b) from Lemma D.2.

### D.1.3. Proof of Lemma C.3

In this section, we prove a full version of Lemma C.3 as stated in Lemma D.3 below. Equation (17a) from Lemma C.3 follows directly as a straightforward corollary.

In Lemma D.3, we consider a general class of distributions parameterized by $\theta$ that models the binary preference $\mathbb{P}_\theta(\vec{\boldsymbol{y}}^a \succ \vec{\boldsymbol{y}}^b \mid x)$. The negative log-likelihood function is defined as

$$\mathcal{L}(\theta) = -\mathbb{E}_{x \sim \rho;\, (\vec{\boldsymbol{y}}^a, \vec{\boldsymbol{y}}^b) \sim \mu(\cdot \mid x)}\left[w(x) \cdot \log \mathbb{P}_\theta(\vec{\boldsymbol{y}}^w \succ \vec{\boldsymbol{y}}^\ell \mid x)\right].$$

The Bradley-Terry (BT) model described in equation (1) and the corresponding loss function $\mathcal{L}(\theta)$ in equation (49) represent a special case of this general framework.

**Lemma D.3** (Gradient of the loss function $\mathcal{L}(\theta)$, full version)**.** *For a general distribution class $\{\mathbb{P}_\theta\}$, the gradient of $\mathcal{L}(\theta)$ with respect to $\theta$ is given by*

$$\nabla_\theta \mathcal{L}(\theta) = -\mathbb{E}_{x \sim \rho;\, (\vec{\boldsymbol{y}}^a, \vec{\boldsymbol{y}}^b) \sim \overline{\mu}(\cdot \mid x)}\left[w(x) \cdot \left\{\mathbb{P}(\vec{\boldsymbol{y}}^a \succ \vec{\boldsymbol{y}}^b \mid x) - \mathbb{P}_\theta(\vec{\boldsymbol{y}}^a \succ \vec{\boldsymbol{y}}^b \mid x)\right\}\right.$$

$$\left. \cdot \frac{\nabla_\theta \mathbb{P}_\theta(\vec{\boldsymbol{y}}^a \succ \vec{\boldsymbol{y}}^b \mid x)}{\mathbb{P}_\theta(\vec{\boldsymbol{y}}^a \succ \vec{\boldsymbol{y}}^b \mid x)\, \mathbb{P}_\theta(\vec{\boldsymbol{y}}^b \succ \vec{\boldsymbol{y}}^a \mid x)}\right], \tag{48a}$$

*where $\overline{\mu}$ is the average distribution defined in equation (17b). Specifically, for the Bradley-Terry (BT) model where*

$$\mathbb{P}_\theta(\vec{\boldsymbol{y}}^a \succ \vec{\boldsymbol{y}}^b \mid x) = \sigma\left(r_\theta(x, \vec{\boldsymbol{y}}^a) - r_\theta(x, \vec{\boldsymbol{y}}^b)\right) = \left\{1 + \left(\frac{(\pi_\theta/\pi_{\text{ref}})(\vec{\boldsymbol{y}}^b \mid x)}{(\pi_\theta/\pi_{\text{ref}})(\vec{\boldsymbol{y}}^a \mid x)}\right)^\beta\right\}^{-1},$$

*the gradient of $\mathcal{L}(\theta)$ becomes*

$$\nabla_\theta \mathcal{L}(\theta) = -\mathbb{E}_{x \sim \rho;\, (\vec{\boldsymbol{y}}^a, \vec{\boldsymbol{y}}^b) \sim \overline{\mu}(\cdot \mid x)}\left[w(x) \cdot \left\{\sigma\left(r^\star(x, \vec{\boldsymbol{y}}^a) - r^\star(x, \vec{\boldsymbol{y}}^b)\right) - \sigma\left(r_\theta(x, \vec{\boldsymbol{y}}^a) - r_\theta(x, \vec{\boldsymbol{y}}^b)\right)\right\}\right.$$

$$\left. \cdot \left\{\nabla_\theta r_\theta(x, \vec{\boldsymbol{y}}^a) - \nabla_\theta r_\theta(x, \vec{\boldsymbol{y}}^b)\right\}\right]. \tag{48b}$$

For notational simplicity, we focus on the proof for the case where the weight function $w(x) = 1$. The results for a general weight function $w(x) > 0$ can be derived in a similar manner.

Recall that the negative log-likelihood function $\mathcal{L}(\theta)$ is defined as

$$\mathcal{L}(\theta) = \mathbb{E}\left[-\log \mathbb{P}_\theta(\vec{\boldsymbol{y}}^w \succ \vec{\boldsymbol{y}}^\ell \mid x)\right].$$

Based on the data generation mechanism, we can expand the expectation in $\mathcal{L}(\theta)$ as

$$\mathcal{L}(\theta) = \mathbb{E}_{x\sim\rho;\,(\vec{\boldsymbol{y}}^a,\vec{\boldsymbol{y}}^b)\sim\mu(\cdot|x)}\left[\mathbb{P}\big(\vec{\boldsymbol{y}}^a \succ \vec{\boldsymbol{y}}^b \mid x\big)\cdot\big\{-\log\mathbb{P}_\theta\big(\vec{\boldsymbol{y}}^a \succ \vec{\boldsymbol{y}}^b \mid x\big)\big\}\right.$$
$$\left.+\,\mathbb{P}\big(\vec{\boldsymbol{y}}^b \succ \vec{\boldsymbol{y}}^a \mid x\big)\cdot\big\{-\log\mathbb{P}_\theta\big(\vec{\boldsymbol{y}}^b \succ \vec{\boldsymbol{y}}^a \mid x\big)\big\}\right]. \tag{49}$$

Notice that we can exchange the roles of $\vec{\boldsymbol{y}}^a$ and $\vec{\boldsymbol{y}}^b$ in the expectation above. This means that we can equivalently express the expectation using the pair $(\vec{\boldsymbol{y}}^b, \vec{\boldsymbol{y}}^a) \sim \mu(\cdot \mid x)$. This symmetry allows us to replace $\mu$ in equation (49) with the average distribution $\bar{\mu}$ as defined in equation (17b).

Next, we take the gradient of the loss function $\mathcal{L}(\theta)$ with respect to the parameter $\theta$ and obtain

$$\nabla_\theta\,\mathcal{L}(\theta) = \mathbb{E}_{x\sim\rho,\,(\vec{\boldsymbol{y}}^a,\vec{\boldsymbol{y}}^b)\sim\bar{\mu}(\cdot|x)}\left[\frac{\mathbb{P}(\vec{\boldsymbol{y}}^a \succ \vec{\boldsymbol{y}}^b \mid x)}{\mathbb{P}_\theta(\vec{\boldsymbol{y}}^a \succ \vec{\boldsymbol{y}}^b \mid x)}\cdot\big\{-\nabla_\theta\,\mathbb{P}_\theta(\vec{\boldsymbol{y}}^a \succ \vec{\boldsymbol{y}}^b \mid x)\big\}\right.$$
$$\left.+\,\frac{\mathbb{P}(\vec{\boldsymbol{y}}^b \succ \vec{\boldsymbol{y}}^a \mid x)}{\mathbb{P}_\theta(\vec{\boldsymbol{y}}^b \succ \vec{\boldsymbol{y}}^a \mid x)}\cdot\big\{-\nabla_\theta\,\mathbb{P}_\theta(\vec{\boldsymbol{y}}^b \succ \vec{\boldsymbol{y}}^a \mid x)\big\}\right].$$

Note that $\mathbb{P}\big(\vec{\boldsymbol{y}}^b \succ \vec{\boldsymbol{y}}^a \mid x\big) = 1 - \mathbb{P}\big(\vec{\boldsymbol{y}}^a \succ \vec{\boldsymbol{y}}^b \mid x\big)$ and $\mathbb{P}_\theta\big(\vec{\boldsymbol{y}}^b \succ \vec{\boldsymbol{y}}^a \mid x\big) = 1 - \mathbb{P}_\theta\big(\vec{\boldsymbol{y}}^a \succ \vec{\boldsymbol{y}}^b \mid x\big)$. Using this, we can rewrite the gradient as

$$\nabla_\theta\,\mathcal{L}(\theta) = \mathbb{E}_{x\sim\rho;\,(\vec{\boldsymbol{y}}^a,\vec{\boldsymbol{y}}^b)\sim\bar{\mu}(\cdot|x)}\left[\left\{\frac{1-\mathbb{P}(\vec{\boldsymbol{y}}^a \succ \vec{\boldsymbol{y}}^b \mid x)}{1-\mathbb{P}_\theta(\vec{\boldsymbol{y}}^a \succ \vec{\boldsymbol{y}}^b \mid x)} - \frac{\mathbb{P}(\vec{\boldsymbol{y}}^a \succ \vec{\boldsymbol{y}}^b \mid x)}{\mathbb{P}_\theta(\vec{\boldsymbol{y}}^a \succ \vec{\boldsymbol{y}}^b \mid x)}\right\}\cdot\nabla_\theta\,\mathbb{P}_\theta\big(\vec{\boldsymbol{y}}^a \succ \vec{\boldsymbol{y}}^b \mid x\big)\right].$$

We simplify the expression further to obtain

$$\nabla_\theta\,\mathcal{L}(\theta) = \mathbb{E}_{x\sim\rho;\,(\vec{\boldsymbol{y}}^a,\vec{\boldsymbol{y}}^b)\sim\bar{\mu}(\cdot|x)}\left[\big\{\mathbb{P}_\theta\big(\vec{\boldsymbol{y}}^a \succ \vec{\boldsymbol{y}}^b \mid x\big) - \mathbb{P}\big(\vec{\boldsymbol{y}}^a \succ \vec{\boldsymbol{y}}^b \mid x\big)\big\}\cdot\frac{\nabla_\theta\,\mathbb{P}_\theta(\vec{\boldsymbol{y}}^a \succ \vec{\boldsymbol{y}}^b \mid x)}{\mathbb{P}_\theta(\vec{\boldsymbol{y}}^a \succ \vec{\boldsymbol{y}}^b \mid x)\,\mathbb{P}_\theta(\vec{\boldsymbol{y}}^b \succ \vec{\boldsymbol{y}}^a \mid x)}\right].$$

This establishes equation (48a) from Lemma C.3.

As for the Bradley-Terry (BT) model, we use the equality

$$\sigma'(z) = \frac{1}{(1+\exp(-z))(1+\exp(z))} = \sigma(z)\,\sigma(-z)\qquad\text{for any } z\in\mathbb{R}$$

to derive the following expression

$$\frac{\nabla_\theta\,\mathbb{P}_\theta(\vec{\boldsymbol{y}}^a \succ \vec{\boldsymbol{y}}^b \mid x)}{\mathbb{P}_\theta(\vec{\boldsymbol{y}}^a \succ \vec{\boldsymbol{y}}^b \mid x)\,\mathbb{P}_\theta(\vec{\boldsymbol{y}}^b \succ \vec{\boldsymbol{y}}^a \mid x)} = \nabla_\theta\,r_\theta(x,\vec{\boldsymbol{y}}^a) - \nabla_\theta\,r_\theta(x,\vec{\boldsymbol{y}}^b). \tag{50}$$

By substituting this gradient expression from equation (50) into equation (48a), we directly obtain equation (48b), thereby completing the proof of Lemma C.3.

## D.2. Proof of Auxiliary Results for Theorem 4.2

In this section, we present the detailed proofs of the supporting lemmas used in the proof of Theorem 4.2. We begin in Appendix D.2.1 by establishing condition (27), which is crucial for the valid application of the master theorem for $Z$-estimators. Following this, in Appendix D.2.2, we compute the Hessian matrix $\nabla_\theta^2\,\mathcal{L}(\theta^\star)$ explicitly. Finally, in Appendix D.2.3, we derive the asymptotic distribution of the gradient $\nabla_\theta\,\widehat{\mathcal{L}}(\theta^\star)$.

### D.2.1. PROOF OF CONDITION (27)

We begin by rewriting the left-hand side of equation (27) as follows:

$$\Delta := \sqrt{n}\left\{\nabla_\theta\,\widehat{\mathcal{L}}(\widehat{\theta}) - \nabla_\theta\,\mathcal{L}(\widehat{\theta})\right\} - \sqrt{n}\left\{\nabla_\theta\,\widehat{\mathcal{L}}(\theta^\star) - \nabla_\theta\,\mathcal{L}(\theta^\star)\right\}$$
$$= \sqrt{n}\left\{\nabla_\theta\,\widehat{\mathcal{L}}(\widehat{\theta}) - \nabla_\theta\,\widehat{\mathcal{L}}(\theta^\star)\right\} - \sqrt{n}\left\{\nabla_\theta\,\mathcal{L}(\widehat{\theta}) - \nabla_\theta\,\mathcal{L}(\theta^\star)\right\}. \tag{51}$$

We then leverage the smoothness properties of the function $r_\theta$, which guarantee the following approximations:

$$\nabla_\theta \widehat{\mathcal{L}}(\widehat{\theta}) - \nabla_\theta \widehat{\mathcal{L}}(\theta^\star) = \nabla_\theta^2 \widehat{\mathcal{L}}(\theta^\star)(\widehat{\theta} - \theta^\star) + o_p(\|\widehat{\theta} - \theta^\star\|_2), \tag{52a}$$

$$\nabla_\theta \mathcal{L}(\widehat{\theta}) - \nabla_\theta \mathcal{L}(\theta^\star) = \nabla_\theta^2 \mathcal{L}(\theta^\star)(\widehat{\theta} - \theta^\star) + o_p(\|\widehat{\theta} - \theta^\star\|_2). \tag{52b}$$

Assuming these equalities (52a) and (52b) hold, we substitute them into equation (51), leading to

$$\begin{aligned} \Delta &= \sqrt{n}\left\{\nabla_\theta^2 \widehat{\mathcal{L}}(\theta^\star)(\widehat{\theta} - \theta^\star) + o_p(\|\widehat{\theta} - \theta^\star\|_2)\right\} - \sqrt{n}\left\{\nabla_\theta^2 \mathcal{L}(\theta^\star)(\widehat{\theta} - \theta^\star) + o_p(\|\widehat{\theta} - \theta^\star\|_2)\right\} \\ &= \sqrt{n}\left\{\nabla_\theta^2 \widehat{\mathcal{L}}(\theta^\star) - \nabla_\theta^2 \mathcal{L}(\theta^\star)\right\}(\widehat{\theta} - \theta^\star) + o_p(1 + \sqrt{n}\|\widehat{\theta} - \theta^\star\|_2). \end{aligned} \tag{53}$$

Using the law of large numbers, we know that $\nabla_\theta^2 \widehat{\mathcal{L}}(\theta^\star) \xrightarrow{P} \nabla_\theta^2 \mathcal{L}(\theta^\star)$, which implies

$$\sqrt{n}\left\{\nabla_\theta^2 \widehat{\mathcal{L}}(\theta^\star) - \nabla_\theta^2 \mathcal{L}(\theta^\star)\right\}(\widehat{\theta} - \theta^\star) = o_p(\sqrt{n}\|\widehat{\theta} - \theta^\star\|_2).$$

Therefore, we conclude that

$$\Delta = o_p(1 + \sqrt{n}\|\widehat{\theta} - \theta^\star\|_2)$$

as claimed in equation (27).

The only remaining task is to establish the validity of equalities (52a) and (52b).

**Proof of Equalities (52a) and (52b):**  We express the loss function $\widehat{\mathcal{L}}(\theta)$ in the form

$$\widehat{\mathcal{L}}(\theta) := \frac{1}{n}\sum_{i=1}^n w(x_i) \cdot \ell_\theta(x_i, \vec{y}_i^w, \vec{y}_i^\ell),$$

where the function $\ell_\theta$ is defined as

$$\ell_\theta(x, \vec{y}_1, \vec{y}_2) = -\log\sigma(r_\theta(x, \vec{y}_1) - r_\theta(x, \vec{y}_2)).$$

We then calculate the gradient $\nabla_\theta \ell_\theta$ and $\nabla_\theta^2 \ell_\theta$ as follows:

$$\nabla_\theta \ell_\theta(x, \vec{y}_1, \vec{y}_2) = \sigma(r_\theta(x, \vec{y}_2) - r_\theta(x, \vec{y}_1)) \cdot \{\nabla_\theta r_\theta(x, \vec{y}_2) - \nabla_\theta r_\theta(x, \vec{y}_1)\} \qquad \text{and}$$

$$\begin{aligned} \nabla_\theta^2 \ell_\theta(x, \vec{y}_1, \vec{y}_2) &= \sigma'(r_\theta(x, \vec{y}_2) - r_\theta(x, \vec{y}_1)) \cdot \{\nabla_\theta r_\theta(x, \vec{y}_2) - \nabla_\theta r_\theta(x, \vec{y}_1)\}\{\nabla_\theta r_\theta(x, \vec{y}_2) - \nabla_\theta r_\theta(x, \vec{y}_1)\}^\top \\ &\quad + \sigma(r_\theta(x, \vec{y}_2) - r_\theta(x, \vec{y}_1)) \cdot \{\nabla_\theta^2 r_\theta(x, \vec{y}_2) - \nabla_\theta^2 r_\theta(x, \vec{y}_1)\}. \end{aligned}$$

When the reward function $r_\theta(x, \vec{y})$, along with its gradient $\nabla_\theta r_\theta(x, \vec{y})$ and Hessian $\nabla_\theta^2 r_\theta(x, \vec{y})$, is uniformly bounded and Lipschitz continuous with respect to $\theta$ for all $(x, \vec{y}) \in \mathcal{X} \times \mathcal{Y}$, it guarantees that the Hessian of the loss function, $\nabla_\theta^2 \ell_\theta$, is also Lipschitz continuous. This holds with some constant $L > 0$ across all $(x, \vec{y}) \in \mathcal{X} \times \mathcal{Y}$, as demonstrated below:

$$\left\|\nabla_\theta^2 \ell_\theta(x, \vec{y}_1, \vec{y}_2) - \nabla_\theta^2 \ell_{\theta^\star}(x, \vec{y}_1, \vec{y}_2)\right\|_2 \leq L \cdot \|\theta - \theta^\star\|_2.$$

From this Lipschitz property, we deduce

$$\left\|\nabla_\theta \ell_\theta(x, \vec{y}_1, \vec{y}_2) - \nabla_\theta \ell_{\theta^\star}(x, \vec{y}_1, \vec{y}_2) - \nabla_\theta^2 \ell_{\theta^\star}(x, \vec{y}_1, \vec{y}_2)(\theta - \theta^\star)\right\|_2 \leq \frac{L}{2} \cdot \|\theta - \theta^\star\|_2^2$$

and further derive

$$\left\|\nabla_\theta \widehat{\mathcal{L}}(\theta) - \nabla_\theta \widehat{\mathcal{L}}(\theta^\star) - \nabla_\theta^2 \widehat{\mathcal{L}}(\theta^\star)(\theta - \theta^\star)\right\|_2 \leq \frac{L\|w\|_\infty}{2} \cdot \|\theta - \theta^\star\|_2^2,$$

$$\left\|\nabla_\theta \mathcal{L}(\theta) - \nabla_\theta \mathcal{L}(\theta^\star) - \nabla_\theta^2 \mathcal{L}(\theta^\star)(\theta - \theta^\star)\right\|_2 \leq \frac{L\|w\|_\infty}{2} \cdot \|\theta - \theta^\star\|_2^2.$$

Finally, under the condition that $\widehat{\theta} \xrightarrow{p} \theta^\star$, these results simplify to the expressions given in equations (52a) and (52b), as previously claimed.

D.2.2. Proof of Lemma C.5, Explicit Form of Hessian $\nabla_\theta^2 \mathcal{L}(\theta^\star)$

From equation (17a) in Lemma C.3, we recall the explicit formula for the gradient $\nabla_\theta \mathcal{L}(\theta)$. Taking the derivative of both sides of equation (17a), we obtain

$$
\begin{aligned}
\nabla_\theta^2 \mathcal{L}(\theta) \;=\; & \mathbb{E}_{x\sim\rho;\,(\vec{\boldsymbol{y}}^a,\vec{\boldsymbol{y}}^b)\sim\overline{\mu}(\cdot|x)}\Big[ w(x)\cdot\sigma'\big(r_\theta(x,\vec{\boldsymbol{y}}^a)-r_\theta(x,\vec{\boldsymbol{y}}^b)\big) \\
& \qquad\qquad \cdot\big\{\nabla_\theta\, r_\theta(x,\vec{\boldsymbol{y}}^a)-\nabla_\theta\, r_\theta(x,\vec{\boldsymbol{y}}^b)\big\}\big\{\nabla_\theta\, r_\theta(x,\vec{\boldsymbol{y}}^a)-\nabla_\theta\, r_\theta(x,\vec{\boldsymbol{y}}^b)\big\}^\top\Big] \\
& -\mathbb{E}_{x\sim\rho;\,(\vec{\boldsymbol{y}}^a,\vec{\boldsymbol{y}}^b)\sim\overline{\mu}(\cdot|x)}\Big[ w(x)\cdot\big\{\sigma\big(r^\star(x,\vec{\boldsymbol{y}}^a)-r^\star(x,\vec{\boldsymbol{y}}^b)\big)-\sigma\big(r_\theta(x,\vec{\boldsymbol{y}}^a)-r_\theta(x,\vec{\boldsymbol{y}}^b)\big)\big\} \\
& \qquad\qquad \cdot\big\{\nabla_\theta^2\, r_\theta(x,\vec{\boldsymbol{y}}^a)-\nabla_\theta^2\, r_\theta(x,\vec{\boldsymbol{y}}^b)\big\}\Big].
\end{aligned}
\tag{54}
$$

When we set $\theta = \theta^\star$, it follows that $r_\theta = r^\star$. This simplification eliminates the second term in expression (54), reducing the Hessian matrix to

$$
\begin{aligned}
\nabla_\theta^2 \mathcal{L}(\theta^\star) \;=\; & \mathbb{E}_{x\sim\rho;\,(\vec{\boldsymbol{y}}^a,\vec{\boldsymbol{y}}^b)\sim\overline{\mu}(\cdot|x)}\Big[ w(x)\cdot\sigma'\big(r^\star(x,\vec{\boldsymbol{y}}^a)-r^\star(x,\vec{\boldsymbol{y}}^b)\big) \\
& \qquad \cdot\big\{\nabla_\theta\, r^\star(x,\vec{\boldsymbol{y}}^a)-\nabla_\theta\, r^\star(x,\vec{\boldsymbol{y}}^b)\big\}\big\{\nabla_\theta\, r^\star(x,\vec{\boldsymbol{y}}^a)-\nabla_\theta\, r^\star(x,\vec{\boldsymbol{y}}^b)\big\}^\top\Big].
\end{aligned}
$$

Substituting the derivative $\sigma'$ with its explicit form, $\sigma'(z) = \sigma(z)\,\sigma(-z)$ for any $z \in \mathbb{R}$, we refine the expression to

$$
\nabla_\theta^2 \mathcal{L}(\theta^\star) \;=\; \Sigma_\star \,,
$$

where the covariance matrix $\Sigma_\star$ is defined in equation (14). This completes the proof of expression (28) from Lemma C.5.

D.2.3. Proof of Lemma C.6, Asymptotic Distribution of Gradient $\nabla_\theta \widehat{\mathcal{L}}(\theta^\star)$

In this section, we analyze the asymptotic distribution of the gradient $\nabla_\theta \widehat{\mathcal{L}}(\theta)$ at $\theta = \theta^\star$, where the loss function $\widehat{\mathcal{L}}(\theta)$ is defined as

$$
\widehat{\mathcal{L}}(\theta) \;=\; -\frac{1}{n}\sum_{i=1}^n w(x)\cdot\log\sigma\Big(r_\theta\big(x_i,\vec{\boldsymbol{y}}_i^w\big)-r_\theta\big(x_i,\vec{\boldsymbol{y}}_i^\ell\big)\Big).
$$

Using the definition of the sigmoid function $\sigma$, we calculate that

$$
(\log\sigma(z))' = \sigma'(z)/\sigma(z) = \sigma(z)\,\sigma(-z)/\sigma(z) = \sigma(-z) \qquad \text{for any real number } z \in \mathbb{R}.
$$

This allows us to reformulate $\nabla_\theta \widehat{\mathcal{L}}(\theta)$ as the average of $n$ i.i.d. vectors $\{\boldsymbol{u}_i\}_{i=1}^n$:

$$
\nabla_\theta \widehat{\mathcal{L}}(\theta) \;=\; \frac{1}{n}\sum_{i=1}^n \boldsymbol{u}_i \,.
\tag{55}
$$

Here each vector $\boldsymbol{u}_i \in \mathbb{R}^d$ is defined as

$$
\boldsymbol{u}_i \;:=\; w(x)\cdot\sigma\big(r_\theta(x_i,\vec{\boldsymbol{y}}_i^\ell)-r_\theta(x_i,\vec{\boldsymbol{y}}_i^w)\big)\cdot\big\{\nabla_\theta\, r_\theta(x_i,\vec{\boldsymbol{y}}_i^\ell)-\nabla_\theta\, r_\theta(x_i,\vec{\boldsymbol{y}}_i^w)\big\}\,.
$$

At $\theta = \theta^\star$, we denote $\boldsymbol{u}_i$ as $\boldsymbol{u}_i^\star$ and $\boldsymbol{g}_i$ as $\boldsymbol{g}_i^\star$. Notably, vector $\boldsymbol{u}_i$ can be rewritten as

$$
\boldsymbol{u}_i \;=\; w(x)\cdot\big\{\sigma\big(r_\theta(x_i,\vec{\boldsymbol{y}}_i^a)-r_\theta(x_i,\vec{\boldsymbol{y}}_i^b)\big)-\mathbb{1}\{\vec{\boldsymbol{y}}_i^a=\vec{\boldsymbol{y}}_i^w,\vec{\boldsymbol{y}}_i^b=\vec{\boldsymbol{y}}_i^\ell\}\big\}\cdot\boldsymbol{g}_i\,,
\tag{56}
$$

where $\boldsymbol{g}_i$ is given by

$$
\boldsymbol{g}_i \;:=\; \nabla_\theta\, r_\theta(x_i,\vec{\boldsymbol{y}}_i^a)-\nabla_\theta\, r_\theta(x_i,\vec{\boldsymbol{y}}_i^b)\,.
$$

From the structure of the BT model, it holds that

$$
\mathbb{E}\big[\mathbb{1}\{\vec{\boldsymbol{y}}_i^a=\vec{\boldsymbol{y}}_i^w,\vec{\boldsymbol{y}}_i^b=\vec{\boldsymbol{y}}_i^\ell\}\mid x_i\big] \;=\; \sigma\big(r^\star(x_i,\vec{\boldsymbol{y}}_i^a)-r^\star(x_i,\vec{\boldsymbol{y}}_i^b)\big)\,,
$$

which implies $\mathbb{E}[\boldsymbol{u}_i^\star] = \boldsymbol{0}$.

To analyze the asymptotic distribution of $\nabla_\theta \widehat{\mathcal{L}}(\theta^\star)$, we apply the central limit theorem (CLT) to its empirical form given in equation (55). By the CLT, we have

$$\sqrt{n}\left(\nabla_\theta \widehat{\mathcal{L}}(\theta^\star) - \nabla_\theta \mathcal{L}(\theta^\star)\right) \xrightarrow{d} \mathcal{N}\left(\boldsymbol{0}, \widetilde{\boldsymbol{\Omega}}\right), \qquad n \to \infty, \tag{57}$$

where the covariance matrix $\widetilde{\boldsymbol{\Omega}} \in \mathbb{R}^{d \times d}$ is given by

$$\widetilde{\boldsymbol{\Omega}} := \operatorname{Cov}(\boldsymbol{u}_1^\star) = \mathbb{E}\left[\boldsymbol{u}_1^\star(\boldsymbol{u}_1^\star)^\top\right].$$

Here we have used the property $\mathbb{E}[\boldsymbol{u}_i^\star] = \boldsymbol{0}$ in the second equality.

We now compute the explicit form of the covariance matrix $\widetilde{\boldsymbol{\Omega}}$. Using the definition of $\boldsymbol{u}_i$ from expression (56), we find that

$$\begin{aligned}
\widetilde{\boldsymbol{\Omega}} &= \mathbb{E}\left[\boldsymbol{u}_1^\star(\boldsymbol{u}_1^\star)^\top\right] \\
&= \mathbb{E}_{x \sim \rho;\, (\vec{\boldsymbol{y}}^a, \vec{\boldsymbol{y}}^b) \sim \overline{\mu}(\cdot|x)}\left[w^2(x) \cdot \left\{\sigma\left(r^\star(x_1, \vec{\boldsymbol{y}}_1^a) - r^\star(x_1, \vec{\boldsymbol{y}}_1^b)\right) - \mathbb{1}\{\vec{\boldsymbol{y}}_1^a = \vec{\boldsymbol{y}}_1^w, \vec{\boldsymbol{y}}_1^b = \vec{\boldsymbol{y}}_1^\ell\}\right\}^2 \cdot \boldsymbol{g}_1^\star(\boldsymbol{g}_1^\star)^\top\right].
\end{aligned} \tag{58}$$

Taking the conditional expectation over the outcomes of winners and losers, and using the relation

$$\begin{aligned}
\mathbb{E}&\left[\left\{\sigma\left(r^\star(x_1, \vec{\boldsymbol{y}}_1^a) - r^\star(x_1, \vec{\boldsymbol{y}}_1^b)\right) - \mathbb{1}\{\vec{\boldsymbol{y}}_1^a = \vec{\boldsymbol{y}}_1^w, \vec{\boldsymbol{y}}_1^b = \vec{\boldsymbol{y}}_1^\ell\}\right\}^2 \;\Big|\; x_1, \vec{\boldsymbol{y}}_1^a, \vec{\boldsymbol{y}}_1^b\right] \\
&= \operatorname{Var}\left(\mathbb{1}\{\vec{\boldsymbol{y}}_1^a = \vec{\boldsymbol{y}}_1^w, \vec{\boldsymbol{y}}_1^b = \vec{\boldsymbol{y}}_1^\ell\} \;\Big|\; x_1, \vec{\boldsymbol{y}}_1^a, \vec{\boldsymbol{y}}_1^b\right) \\
&= \sigma\left(r^\star(x_i, \vec{\boldsymbol{y}}_i^a) - r^\star(x_i, \vec{\boldsymbol{y}}_i^b)\right)\sigma\left(r^\star(x_i, \vec{\boldsymbol{y}}_i^b) - r^\star(x_i, \vec{\boldsymbol{y}}_i^a)\right),
\end{aligned}$$

we reduce equation (58) to

$$\widetilde{\boldsymbol{\Omega}} = \mathbb{E}_{x \sim \rho;\, (\vec{\boldsymbol{y}}^a, \vec{\boldsymbol{y}}^b) \sim \overline{\mu}(\cdot|x)}\left[w^2(x) \cdot \operatorname{Var}\left(\mathbb{1}\{\vec{\boldsymbol{y}}_1^a = \vec{\boldsymbol{y}}_1^w, \vec{\boldsymbol{y}}_1^b = \vec{\boldsymbol{y}}_1^\ell\} \mid x_1, \vec{\boldsymbol{y}}_1^a, \vec{\boldsymbol{y}}_1^b\right) \cdot \boldsymbol{g}_1^\star(\boldsymbol{g}_1^\star)^\top\right].$$

Bounding the weight function $w(x)$ by its uniform bound $\|w\|_\infty$, we simplify further:

$$\widetilde{\boldsymbol{\Omega}} \preceq \|w\|_\infty \cdot \mathbb{E}\left[w(x) \cdot \operatorname{Var}\left(\mathbb{1}\{\vec{\boldsymbol{y}}_1^a = \vec{\boldsymbol{y}}_1^w, \vec{\boldsymbol{y}}_1^b = \vec{\boldsymbol{y}}_1^\ell\} \mid x_1, \vec{\boldsymbol{y}}_1^a, \vec{\boldsymbol{y}}_1^b\right) \cdot \boldsymbol{g}_1^\star(\boldsymbol{g}_1^\star)^\top\right].$$

This ultimately reduces to

$$\widetilde{\boldsymbol{\Omega}} \preceq \|w\|_\infty \cdot \boldsymbol{\Sigma}_\star \tag{59}$$

where $\boldsymbol{\Sigma}_\star$ is defined in equation (14).

Finally, by combining equations (57) and (59), we establish the asymptotic normality of $\nabla_\theta \widehat{\mathcal{L}}(\theta^\star)$ and complete the proof of Lemma C.6.

### D.3. Proof of Auxiliary Results for Theorem 4.3

This section contains the proofs of the auxiliary results supporting Theorem 4.3. In Appendix D.3.1, we derive the explicit form of the Hessian $\nabla_\theta^2 J(\pi^\star)$. Appendix D.3.2 rigorously establishes the asymptotic distribution of the value gap (equation (34)). Finally, Appendix D.3.3 proves the tail bound (13) on the chi-square distribution $\chi_d^2$.

### D.3.1. PROOF OF EQUATION (12) FROM THEOREM 4.3, EXPLICIT FORM OF HESSIAN $\nabla_\theta^2 J(\pi^\star)$

We begin by differentiating expression (41) for the gradient $\nabla_\theta J(\pi_\theta)$ to obtain the Hessian matrix $\nabla_\theta^2 J(\pi_\theta)$. The resulting expression can be written as

$$\nabla_\theta^2 J(\pi_\theta) = \boldsymbol{\Gamma}_1 + \boldsymbol{\Gamma}_2 + \boldsymbol{\Gamma}_3,$$

where the terms are defined as follows:

$$\boldsymbol{\Gamma}_1 := \frac{1}{\beta}\, \mathbb{E}_{x\sim\rho}\left[\int_{\mathcal{Y}}\left\{r^\star(x,\vec{\boldsymbol{y}}) - r_\theta(x,\vec{\boldsymbol{y}})\right\}\left\{\nabla_\theta\, r_\theta(x,\vec{\boldsymbol{y}}) - \mathbb{E}_{\vec{\boldsymbol{y}}'\sim\pi_\theta(\cdot|x)}\left[\nabla_\theta\, r_\theta(x,\vec{\boldsymbol{y}}')\right]\right\}\nabla_\theta\,\pi_\theta(d\vec{\boldsymbol{y}}\mid x)^\top\right],$$

$$\boldsymbol{\Gamma}_2 := -\frac{1}{\beta}\, \mathbb{E}_{x\sim\rho,\,\vec{\boldsymbol{y}}\sim\pi_\theta(\cdot|x)}\left[\left\{\nabla_\theta\, r_\theta(x,\vec{\boldsymbol{y}}) - \mathbb{E}_{\vec{\boldsymbol{y}}'\sim\pi_\theta(\cdot|x)}\left[\nabla_\theta\, r_\theta(x,\vec{\boldsymbol{y}}')\right]\right\}\nabla_\theta\, r_\theta(x,\vec{\boldsymbol{y}})^\top\right],$$

$$\boldsymbol{\Gamma}_3 := \frac{1}{\beta}\, \mathbb{E}_{x\sim\rho,\,\vec{\boldsymbol{y}}\sim\pi_\theta(\cdot|x)}\left[\left\{r^\star(x,\vec{\boldsymbol{y}}) - r_\theta(x,\vec{\boldsymbol{y}})\right\}\left\{\nabla_\theta^2\, r_\theta(x,\vec{\boldsymbol{y}}) - \nabla_\theta\, \mathbb{E}_{\vec{\boldsymbol{y}}'\sim\pi_\theta(\cdot|x)}\left[\nabla_\theta\, r_\theta(x,\vec{\boldsymbol{y}}')\right]\right\}\right].$$

At the point $\theta = \theta^\star$, we know that $r_\theta = r^\star$. This simplifies the expression significantly:

$$\boldsymbol{\Gamma}_1 = \boldsymbol{0} \quad \text{and} \quad \boldsymbol{\Gamma}_3 = \boldsymbol{0}.$$

Therefore, only term $\boldsymbol{\Gamma}_2$ contributes to the Hessian, and it further reduces to

$$\begin{aligned}
\boldsymbol{\Gamma}_2 &= -\frac{1}{\beta}\, \mathbb{E}_{x\sim\rho,\,\vec{\boldsymbol{y}}\sim\pi_\theta(\cdot|x)}\left[\nabla_\theta\, r_\theta(x,\vec{\boldsymbol{y}})\,\nabla_\theta\, r_\theta(x,\vec{\boldsymbol{y}})^\top\right]\\
&\quad + \frac{1}{\beta}\, \mathbb{E}_{x\sim\rho}\left[\mathbb{E}_{\vec{\boldsymbol{y}}'\sim\pi_\theta(\cdot|x)}\left[\nabla_\theta\, r_\theta(x,\vec{\boldsymbol{y}}')\right]\mathbb{E}_{\vec{\boldsymbol{y}}\sim\pi_\theta(\cdot|x)}\left[\nabla_\theta\, r_\theta(x,\vec{\boldsymbol{y}})\right]^\top\right]\\
&= -\frac{1}{\beta}\, \mathbb{E}_{x\sim\rho}\left[\mathrm{Cov}_{\vec{\boldsymbol{y}}\sim\pi_\theta(\cdot|x)}\left[\nabla_\theta\, r_\theta(x,\vec{\boldsymbol{y}})\mid x\right]\right].
\end{aligned}$$

From this simplification, we deduce

$$\nabla_\theta^2\, J(\pi^\star) = -\frac{1}{\beta}\, \mathbb{E}_{x\sim\rho}\left[\mathrm{Cov}_{\vec{\boldsymbol{y}}\sim\pi^\star(\cdot|x)}\left[\nabla_\theta\, r^\star(x,\vec{\boldsymbol{y}})\mid x\right]\right],$$

which establishes equation (12) as stated in Theorem 4.3.

### D.3.2. PROOF OF THE ASYMPTOTIC DISTRIBUTION IN EQUATION (34)

The goal of this part is to establish the asymptotic distribution of $n\{J(\pi^\star) - J(\widehat{\pi})\}$, as stated in equation (34) from Appendix C.2.3. To achieve this, we first recast the value gap into the product of two terms and then invoke Slutsky's theorem.

We start by writing

$$n\cdot\{J(\pi^\star) - J(\widehat{\pi})\} = \underbrace{n\cdot(\widehat{\theta} - \theta^\star)^\top \boldsymbol{H}\,(\widehat{\theta} - \theta^\star)}_{U_n}\cdot\underbrace{\frac{J(\pi^\star) - J(\widehat{\pi})}{(\widehat{\theta} - \theta^\star)^\top \boldsymbol{H}\,(\widehat{\theta} - \theta^\star)}}_{V_n}. \tag{60}$$

By isolating $U_n$ and $V_n$ in this way, we can handle their limiting behaviors separately:

$$U_n \xrightarrow{d} \boldsymbol{z}^\top \boldsymbol{\Omega}^{\frac{1}{2}} \boldsymbol{H}\boldsymbol{\Omega}^{\frac{1}{2}}\boldsymbol{z} \qquad \text{with } \boldsymbol{z}\sim\mathcal{N}(\boldsymbol{0},\boldsymbol{I}), \tag{61a}$$

$$V_n \xrightarrow{p} \frac{1}{2}. \tag{61b}$$

If these two results are established, the desired asymptotic distribution of the value gap, as given in equation (34), follows directly from Slutsky's theorem.

To complete the proof, we proceed to verify equations (61a) and (61b). It is worth noting that equation (61a) is a straightforward corollary of Theorem 4.2, so the main task is to establish the convergence result in equation (61b).

**Proof of Equation** (61b): Since $\boldsymbol{\Sigma}_\star$ is nonsingular, the matrix $\boldsymbol{H} = (\overline{Z}_\theta/\beta)\cdot\boldsymbol{\Sigma}_\star$ is also nonsingular. From equation (33), we know that for any $\varepsilon\in(0,1)$, there exists a threshold $\eta(\varepsilon) > 0$ such that whenever $\|\theta - \theta^\star\|_2 \leq \eta(\varepsilon)$, the following inequality holds:

$$\left(\frac{1}{2} - \varepsilon\right)(\theta - \theta^\star)^\top \boldsymbol{H}\,(\theta - \theta^\star) \leq J(\pi^\star) - J(\pi_\theta) \leq \left(\frac{1}{2} + \varepsilon\right)(\theta - \theta^\star)^\top \boldsymbol{H}\,(\theta - \theta^\star).$$

This can be reformulated as

$$\left| V_n - \frac{1}{2} \right| \leq \varepsilon.$$

Next, under the condition that $\widehat{\theta} \xrightarrow{p} \theta^\star$, for any $\delta > 0$, there exists an integer $N(\varepsilon, \delta) \in \mathbb{Z}_+$ such that for any $n \geq N(\varepsilon, \delta)$,

$$\mathbb{P}\{\|\widehat{\theta} - \theta^\star\|_2 > \eta(\varepsilon)\} \leq \delta.$$

Therefore, for any $n \geq N(\varepsilon, \delta)$, we can conclude

$$\mathbb{P}\left\{ \left| V_n - \frac{1}{2} \right| > \varepsilon \right\} \leq \delta.$$

In simpler terms, $V_n \xrightarrow{p} \frac{1}{2}$, which establishes equation (61b).

### D.3.3. PROOF OF THE TAIL BOUND IN EQUATION (13)

We now establish the tail bound

$$\mathbb{P}\{\chi_d^2 > (1+\varepsilon)\, d\} \leq \exp\left\{ -\frac{d}{2}\big(\varepsilon - \log(1+\varepsilon)\big) \right\}, \tag{62}$$

as stated in equation (13).

We first note that the moment-generating function (MGF) of distribution $\chi_d^2$ is given by

$$M_{\chi_d^2}(t) = (1 - 2t)^{-\frac{d}{2}}, \quad \text{for any } t < \tfrac{1}{2}.$$

Using Markov's inequality, for any $t > 0$, we have

$$\mathbb{P}\{\chi_d^2 > (1+\varepsilon)\, d\} \leq \exp\{-t(1+\varepsilon)d\} \cdot M_{\chi_d^2}(t) = \exp\{-t(1+\varepsilon)d\} \cdot (1-2t)^{-\frac{d}{2}}, \qquad \text{for any } t < \tfrac{1}{2}. \tag{63}$$

We optimize the bound by choosing $t$ to minimize the exponent $-t(1+\varepsilon)d - \frac{d}{2}\log(1-2t)$. Solving for the optimal $t$, we obtain

$$t = \frac{\varepsilon}{2(1+\varepsilon)}.$$

Substituting $t$ back into inequality (63), the bound simplifies to the desired inequality (62).

### D.4. Proof of the next-token version of PILAF sampling

**Proof of the Explicit Forms of $\pi_\theta^+(\cdot \mid x, y_{1:t-1})$ and $\pi_\theta^-(\cdot \mid x, y_{1:t-1})$:** To begin, we express the policies $\pi_\theta$ and $\pi_{\text{ref}}$ in terms of their logits. Specifically, each policy can be written in the exponential family form, normalized by a partition function:

$$\pi_\theta(y_t \mid x, y_{1:t-1}) = C_\theta^{-1} \exp\left\{ \boldsymbol{h}_\theta(y_t \mid x, y_{1:t-1}) \right\} \qquad \text{where } C_\theta := \sum_{y \in \mathcal{V}} \exp\left\{ \boldsymbol{h}_\theta(y \mid x, y_{1:t-1}) \right\};$$

$$\pi_{\text{ref}}(y_t \mid x, y_{1:t-1}) = C_{\text{ref}}^{-1} \exp\left\{ \boldsymbol{h}_{\text{ref}}(y_t \mid x, y_{1:t-1}) \right\} \qquad \text{where } C_{\text{ref}} := \sum_{y \in \mathcal{V}} \exp\left\{ \boldsymbol{h}_{\text{ref}}(y \mid x, y_{1:t-1}) \right\}.$$

Now, let us consider the modified policy $\pi_\theta^+(y_t \mid x, y_{1:t-1})$. Substituting the expressions for $\pi_\theta$ and $\pi_{\text{ref}}$ into its definition, we get

$$\pi_\theta^+(y_t \mid x, y_{1:t-1}) = \frac{1}{Z(x, y_{1:t-1})} \pi_\theta(y_t \mid x, y_{1:t-1}) \left( \frac{\pi_\theta(y_t \mid x, y_{1:t-1})}{\pi_{\text{ref}}(y_t \mid x, y_{1:t-1})} \right)^\beta$$

$$= \frac{1}{Z(x, y_{1:t-1})} C_\theta^{-1} \exp\left\{ \boldsymbol{h}_\theta(y_t \mid x, y_{1:t-1}) \right\} \left( \frac{C_\theta^{-1} \exp\{\boldsymbol{h}_\theta(y_t \mid x, y_{1:t-1})\}}{C_{\text{ref}}^{-1} \exp\{\boldsymbol{h}_{\text{ref}}(y_t \mid x, y_{1:t-1})\}} \right)^\beta.$$

At this point, we observe that the terms $Z(x, y_{1:t-1})$, $C_\theta$, and $C_{\text{ref}}$ do not depend on token $y_t$, given the prompt $x$ and previous tokens $y_{1:t-1}$. Therefore, we can treat them as constants when focusing on the structure of $\pi_\theta^+$ as probabilities over token $y_t$. This allows us to simplify the expression:

$$\pi_\theta^+(y_t \mid x, y_{1:t-1}) \propto \frac{\exp\{\boldsymbol{h}_\theta(y_t \mid x, y_{1:t-1})\}^{1+\beta}}{\exp\{\boldsymbol{h}_{\text{ref}}(y_t \mid x, y_{1:t-1})\}^\beta} = \exp\left(\{(1+\beta)\boldsymbol{h}_\theta - \beta\boldsymbol{h}_{\text{ref}}\}(y_t \mid x, y_{1:t-1})\right).$$

It shows that the modified policy $\pi_\theta^+$ is proportional to the exponential of a linear combination of logits from $\pi_\theta$ and $\pi_{\text{ref}}$. To convert this into a proper probability distribution, we normalize over the token space:

$$\pi_\theta^+(y_t \mid x, y_{1:t-1}) = \frac{\exp\left(\{(1+\beta)\boldsymbol{h}_\theta - \beta\boldsymbol{h}_{\text{ref}}\}(y_t \mid x, y_{1:t-1})\right)}{\sum_{y \in \mathcal{V}} \exp\left(\{(1+\beta)\boldsymbol{h}_\theta - \beta\boldsymbol{h}_{\text{ref}}\}(y \mid x, y_{1:t-1})\right)}.$$

In other words, the new policy is simply the softmax over the combined logit:

$$\pi_\theta^+(\cdot \mid x, y_{1:t-1}) = \text{softmax}\left(\{(1+\beta)\boldsymbol{h}_\theta - \beta\boldsymbol{h}_{\text{ref}}\}(x, y_{1:t-1})\right)$$

as claimed.

The expression for the negatively weighted policy $\pi_\theta^-$ can be derived in a similar manner to $\pi_\theta^+$.

# E. Supporting Theorem: Master Theorem for $Z$-Estimators

In this section, we provide a brief introduction to the master theorem for $Z$-estimators for the convenience of the readers.

Let the parameter space be $\Theta$, and consider a data-dependent function $\Psi_n : \Theta \to \mathbb{L}$, where $\mathbb{L}$ is a metric space with norm $\|\cdot\|_{\mathbb{L}}$. Assume that the parameter estimate $\widehat{\theta}_n \in \Theta$ satisfies $\|\Psi_n(\widehat{\theta}_n)\|_{\mathbb{L}} \overset{p}{\to} 0$, making $\widehat{\theta}_n$ a $Z$-estimator. The function $\Psi_n$ is an estimator of a fixed function $\Psi : \Theta \to \mathbb{L}$, where $\Psi(\theta_0) = 0$ for some parameter of interest $\theta_0 \in \Theta$.

**Theorem E.1** (Theorem 2.11 in Kosorok (2008), master theorem for $Z$-estimators). *Suppose the following conditions hold:*

1. *$\Psi(\theta_0) = 0$, where $\theta_0$ lies in the interior of $\Theta$.*

2. *$\sqrt{n}\,\Psi_n(\widehat{\theta}_n) \overset{p}{\to} 0$ and $\|\widehat{\theta}_n - \theta_0\| \overset{p}{\to} 0$ for the sequence of estimators $\{\widehat{\theta}_n\} \subset \Theta$.*

3. *$\sqrt{n}(\Psi_n - \Psi)(\theta_0) \overset{d}{\to} Z$, where $Z$ is a tight[4] random variable.*

4. *The following smoothness condition is satisfied:*

$$\frac{\left\|\sqrt{n}\big(\Psi_n(\widehat{\theta}_n) - \Psi(\widehat{\theta}_n)\big) - \sqrt{n}\big(\Psi_n(\theta_0) - \Psi(\theta_0)\big)\right\|_{\mathbb{L}}}{1 + \sqrt{n}\,\|\widehat{\theta}_n - \theta_0\|} \overset{p}{\to} 0. \tag{64}$$

*Additionally, assume that $\theta \mapsto \Psi(\theta)$ is Fréchet differentiable[5] at $\theta_0$ with derivative $\dot{\Psi}_{\theta_0}$, and that $\dot{\Psi}_{\theta_0}$ is continuously invertible[6]. Then*

$$\left\|\sqrt{n}\dot{\Psi}_{\theta_0}(\widehat{\theta}_n - \theta_0) + \sqrt{n}(\Psi_n - \Psi)(\theta_0)\right\|_{\mathbb{L}} \overset{p}{\to} 0$$

*and therefore*

$$\sqrt{n}\left(\widehat{\theta}_n - \theta_0\right) \overset{d}{\to} -\dot{\Psi}_{\theta_0}^{-1} Z.$$

---

[4]A random variable $Z$ is tight if, for any $\epsilon > 0$, there exists a compact set $K \subset \mathbb{R}$ such that $\mathbb{P}(Z \notin K) < \epsilon$.

[5]Fréchet differentiability: A map $\phi : \mathbb{D} \to \mathbb{L}$ is Fréchet differentiable at $\theta$ if there exists a continuous, linear map $\phi'_\theta : \mathbb{D} \to \mathbb{L}$ such that $\|\phi(\theta + h_n) - \phi(\theta) - \phi'_\theta(h_n)\|_{\mathbb{L}}/\|h_n\| \to 0$ for all sequences $\{h_n\} \subset \mathbb{D}$ with $\|h_n\| \to 0$ and $\theta + h_n \in \Theta$ for all $n \geq 1$.

[6]Continuous invertibility: A map $A : \Theta \to \mathbb{L}$ is continuously invertible if $A$ is invertible, and there exists a constant $c > 0$ such that $\|A(\theta_1) - A(\theta_2)\|_{\mathbb{L}} \geq c\|\theta_1 - \theta_2\|$ for all $\theta_1, \theta_2 \in \Theta$.

# F. Experimental Details

We implement our code based on the open-sourced OpenRLHF framework Hu et al. (2024). We will open-source our code in the camera-ready version.

We use both the helpful and the harmless (HH) sets from HH-RLHF (Bai et al., 2022) without additional data selection. We adopt the chat template from the Skywork-Reward-8B model (Liu et al., 2024a) to align with the reward template. This reward model, fine-tuned from Llama-3.1-8B, is used to simulate human preference labeling and matches our network trained for alignment.

For SFT, we apply full-parameter tuning with Adam for one epoch, using a cosine learning rate schedule, a 3% warmup phase, a learning rate of $5 \times 10^{-7}$, and a batch size of 256. These hyperparameters are adopted from Hu et al. (2024).

For all the DPO training in both iterative and online settings, we use full-parameter tuning with Adam but with two epochs. The learning rate, warmup schedules, and batch size are all the same.

During generation, we limit the maximum number of new tokens to 896 and employ top_p decoding with $p = 0.95$ for all experiments. For Online DPO, we use a sampling temperature of 1.0, following Guo et al. (2024), while in Iterative DPO, we set the temperature to 0.7 to account for the off-policy nature of the data, following Dong et al. (2024); Shi et al. (2024).

Prompts are truncated to a maximum length of 512 tokens (truncated from the left if the length exceeds this limit) for SFT, DPO, and generation tasks. For SFT data, the maximum length is further restricted to 1024 tokens. When the combined length of the response and the (truncated) prompt exceeds 1024 tokens, the response is truncated from the right. These truncation practices align with the standard methodology described by Rafailov et al. (2023). In contrast, for DPO, responses are not further truncated, as we are already limiting the maximum tokens generated during the generation process.

When reproducing the *Hybrid Sampling* baseline (Exploration Preference Optimization, XPO) from Xie et al. (2024), we use $\alpha = 5 \times 10^{-6}$ as suggested in the paper.

We do not include a comparison with Shi et al. (2024) and Liu et al. (2024c) in our experiments. While Shi et al. (2024) employs a sampling method similar to ours, their approach requires significantly more hyperparameters to tune, whereas our method involves no hyperparameter tuning. On the other hand, Liu et al. (2024c) relies on training an ensemble of 20 reward models to approximate the posterior. Their sampling method requires solving the argmax of these rewards, which is computationally intractable. As a workaround, they generate 20 samples and select the best one using best-of-N with $N = 20$. This approach demands at least six times the computational resources compared to our method.

## F.1. Additional Results

We present the full results for Online DPO with the overfitted initial policy, including a scatter plot in Figure 6 and a summary of the objective values in Table 4.

We observe that *Vanilla Sampling* rapidly increases its KL divergence from the reference model while its reward improvement diminishes over time. In contrast, PILAF undergoes an early phase of training with fluctuating KL values but ultimately achieves a policy with higher reward and substantially lower KL divergence. We hypothesize that PILAF's interpolation-based exploration enables it to escape the suboptimal region of the loss landscape where *Vanilla Sampling* remains trapped.

Conversely, *Hybrid Sampling*, despite its explicit exploration design, remains biased by the policy model and continues to exhibit high KL values. While KL divergence decreases over training, the reward improvement remains limited. Meanwhile, *Best-of-N Sampling* introduces an implicit exploration mechanism through internal DPO, which selects the best and worst responses, leading to wider coverage than *Vanilla Sampling*. However, despite achieving a KL divergence similar to PILAF, it results in a lower reward. These findings highlight the superiority of PILAF sampling, demonstrating its effectiveness in robustly optimizing an overfitted policy.

# G. Extension to Proximal Policy Optimization (PPO)

In this section, we briefly explore how the core principles of our PILAF sampling approach can be extended to PPO-based RLHF methods.

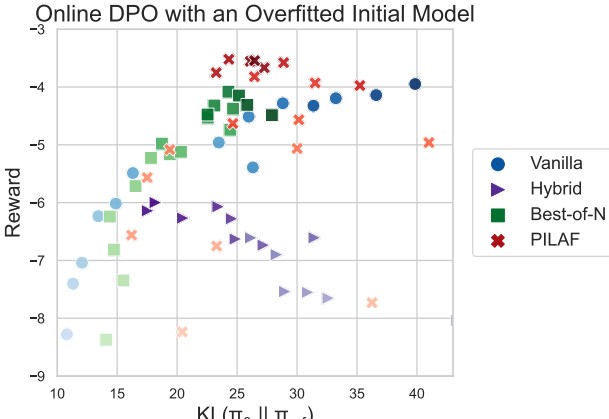

Online DPO with an Overfitted Initial Model

*Table 4.* **Results of Online DPO with an overfitted initial policy.** We report the average reward, KL divergence from the reference model, and objective $J$ on the testset.

| METHOD | REWARD ($\uparrow$) | KL ($\downarrow$) | $J$ ($\uparrow$) |
|---|---|---|---|
| *Vanilla* | -3.95 | 39.85 | -7.93 |
| *Best-of-N* | -4.49 | 27.90 | -7.28 |
| *Hybrid* | -6.00 | **18.20** | -7.82 |
| *PILAF* (OURS) | **-3.54** | 26.45 | **-6.19** |

*Figure 6.* **Online DPO with an overfitted initial policy**. Full results of the Figure 4. Each dot represents an evaluation performed every 50 training steps. Color saturation indicates the training step, with darker colors representing later steps.

**Integrating Response Sampling in InstructGPT:**  The PPO-based RLHF pipeline used in InstructGPT (Ouyang et al., 2022) consists of three key steps:

(i)  Supervised Fine-Tuning (SFT) that produces the reference model $\pi_{\text{ref}}$.
(ii)  Reward Modeling (RM) by solving the optimization problem (2), yielding an estimated reward function $r_\theta$.
(iii)  Reinforcement Learning Fine-Tuning, where the policy $\pi_\phi$ is optimized against the reward model $r_\theta$ using the Proximal Policy Optimization (PPO) algorithm, following the optimization scheme (4).

The key distinction between the PPO and DPO approaches lies in how the reward model $r_\theta$ is represented—explicitly in PPO and implicitly in DPO. In response sampling for data collection, it is crucial to consider the iterative nature of the InstructGPT pipeline. During each iteration, additional human-labeled data is collected for reward modeling (step (ii)), and steps (ii) and (iii) are repeatedly applied to refine the model. Our proposed PILAF algorithm naturally integrates into this pipeline by improving the data collection process in step (ii), thereby enhancing reward model training and, in turn, policy optimization.

**Extensions of T-PILAF and PILAF:**  Extending our response sampling methods, PILAF and T-PILAF, to the PPO setup with an explicit $r_\theta$ is both natural and straightforward.

- Within the theoretical framework of T-PILAF, as introduced in Section 3, the only required modification is replacing $\pi_\theta$ with the language model $\pi_\phi$ and redefining the interpolated and extrapolated policies, $\pi_\phi^+$ and $\pi_\phi^-$, following the same formulation as in equations (9a) and (9b). Specifically, we define

$$\pi_\phi^+(\vec{\boldsymbol{y}} \mid x) := \frac{1}{Z^+(x)}\, \pi_\phi(\vec{\boldsymbol{y}} \mid x) \exp\left\{ r_\theta(x, \vec{\boldsymbol{y}}) \right\}, \tag{65a}$$

$$\pi_\phi^-(\vec{\boldsymbol{y}} \mid x) := \frac{1}{Z^-(x)}\, \pi_\phi(\vec{\boldsymbol{y}} \mid x) \exp\left\{ - r_\theta(x, \vec{\boldsymbol{y}}) \right\}, \tag{65b}$$

where $r_\theta$ is now explicitly produced by a reward network, rather than being implicitly derived from $\pi_\phi$, as in equation (5).

- To extend our empirical PILAF algorithm, as described in Section 5, we propose applying the same interpolation and extrapolation techniques directly to the logits of the language models $\pi_\phi$ and $\pi_{\text{ref}}$. In particular, we take

$$\pi_\phi^+(\cdot \mid x, y_{1:t-1}) = \mathsf{softmax}\Big(\left\{(1 + \beta)\, \boldsymbol{h}_\phi - \beta\, \boldsymbol{h}_{\text{ref}}\right\}(x, y_{1:t-1})\Big),$$

$$\pi_\phi^-(\cdot \mid x, y_{1:t-1}) = \mathsf{softmax}\Big(\left\{(1 - \beta)\, \boldsymbol{h}_\phi + \beta\, \boldsymbol{h}_{\text{ref}}\right\}(x, y_{1:t-1})\Big),$$

where $\boldsymbol{h}_\phi$ and $\boldsymbol{h}_{\text{ref}}$ represent the logits of the language models $\pi_\phi$ and $\pi_{\text{ref}}$, respectively.

**Adaption of Theoretical Analysis:** Our theoretical analyses can be extended to the PPO framework, assuming that the optimization process (4) in step (iii) of InstructGPT is solved exactly. In this case, the policy satisfies $\pi_\phi = \pi_{r_\theta}$, where

$$\pi_{r_\theta}(\vec{\boldsymbol{y}} \mid x) \ := \ \frac{1}{Z_\theta(x)} \, \pi_{\text{ref}}(\vec{\boldsymbol{y}} \mid x) \exp\left\{ \frac{1}{\beta} \, r_\theta(x, \vec{\boldsymbol{y}}) \right\}.$$

Under this assumption, the output language model $\pi_\phi$ is implicitly a function of the parameter $\theta$. Building on this, we can adapt our optimization and statistical analyses as follows:

- **Optimization Consideration:** Using the same argument as in Theorem 4.1, we can prove that

$$\nabla_\theta \, \mathcal{L}(\theta) \ = \ -\, C' \cdot \nabla_\theta \, J(\pi_\phi) \ + \ T_2 \,,$$

  where $C' > 0$ is a universal constant, and $T_2$ represents a second-order approximation error.

  In other words, if the policy optimization step is sufficiently accurate for the reward model $r_\theta$, then performing gradient descent on the MLE loss with respect to $\theta$ is equivalent to applying gradient ascent on the oracle objective $J$, following the steepest direction in the parameter space of $\theta$.

- **Statistical Consideration:** Even with the new parameterization, the asymptotic distribution of $\widehat{\theta}$ from Theorem 4.2 remains unchanged. Moreover, the gradient and Hessian of $J$ with respect to $\theta$ retain the same form as in Theorem 4.1. As a result, the statistical analysis extends naturally to PPO, allowing us to conclude that PILAF also maintains structure-invariant statistical efficiency for PPO methods.

