# OpenReview forum: "PILAF: Optimal Human Preference Sampling for Reward Modeling"
_ICML.cc/2025/Conference — ICML 2025 poster_

### Official Review · Reviewer_kgC8 · 2025-03-11

**Overall Recommendation:** 3

**Summary:**

This paper introduces PILAF (Policy-Interpolated Learning for Aligned Feedback), a novel sampling strategy for iterative/online DPO. The authors show that with this new sampling algorithm, the gradient of the loss function matches the KL-regularized objective function, and they further provide asymptotic analysis of DPO with PILAF. For the experiments, they implement PILAF with some empirical approximation and evaluate it both with iterative and online DPO, where it consistently outperforms baseline methods (XPO and Best-of-N).

**Claims And Evidence:**

Yes.

**Essential References Not Discussed:**

"Value-Incentivized Preference Optimization: A Unified Approach to Online and Offline RLHF" by Cen et al. also proposes a variant of DPO, which has similar structures to XPO. The authors should also add this work into discussion.

**Experimental Designs Or Analyses:**

The experiments look good to me. However, I suggest that the authors add VPO in "Value-Incentivized Preference Optimization: A Unified Approach to Online and Offline RLHF" by Cen et al. to the baselines because it can also be considered as a variant of DPO.

**Methods And Evaluation Criteria:**

Yes.

**Other Comments Or Suggestions:**

None.

**Other Strengths And Weaknesses:**

None.

**Questions For Authors:**

None.

**Relation To Broader Scientific Literature:**

The key contribution is the proposed new sampling algorithm for DPO, which has better empirical performance than XPO and best-of-N on the HH-RLHF dataset. However, this contribution is limited because it is specifically designed for DPO, whose performance is not SOTA on most benchmarks. The authors can try whether this sampling strategy is also useful in other algorithms like IPO, KPO and PPO. If this sampling strategy is shown to be useful universally, this work would have a greater impact.

**Theoretical Claims:**

I checked Theorem 4.1~4.3. They look good to me.

---

> ### Author Rebuttal · Authors · 2025-04-01
>
> We thank the reviewer for their thoughtful review, particularly for recognizing the strength of both the theoretical and experimental parts, and for checking the proofs of all theoretical results.
>
> 1. > "Value-Incentivized Preference Optimization". The authors should also add this work into discussion.
>
> We thank the reviewer for bringing this up. Please note that we did cite Cen et al. around line 621. As we discussed there, their method does not modify the sampling scheme; rather, it adds a regularization term to the DPO objective to encourage departure from the calibration samples. Therefore, we keep this reference in the appendix and focus on those papers - more directly comparable to our work -  that modify the sampling scheme in the main body of our paper. We can relocate this distinction in related works to the main body of the paper if you would consider that it adds clarity to our exposition.
>
> 2. > Add VPO to the baselines.
>
> At the reviewer’s suggestion, we have added VPO as a baseline in the online setting. Due to the need to generate responses in all experiments, completing the full VPO comparison is infeasible within the rebuttal period. Therefore, we are adding this baseline for the online setting and plan to complete the comparison for the iterative setting in the camera-ready version. We report the figure at [anonymous URL](https://drive.google.com/file/d/11wfcQT07dxy0IpXuIcdRUrtRBoPPuwXD). The results show that PILAF outperforms VPO, even after a small hyperparameter search for VPO (details can be found in the provided link).
>
> 3. > However, this contribution is limited because it is specifically designed for DPO, whose performance is not SOTA on most benchmarks. The authors can try whether this sampling strategy is also useful in other algorithms like IPO, KPO and PPO.
>
> We would like to clarify that our motivation—and the scope of our theoretical and experimental contributions—is not limited to DPO, but rather to reveal and address a fundamental misalignment that pervades the two‑phase RLHF framework. A prevailing assumption in recent RLHF work is that on‑policy data generated during training constitutes “good” alignment data. However, we demonstrate that—even when using on‑policy samples—the alignment process remains suboptimal.
>
> Specifically, RLHF consists of two sequential phases: first, preference data are collected and used to extract human values and train a reward model— via maximum‑likelihood estimation either explicitly (as in PPO) or implicitly (as in DPO, IPO, and KPO)—and second, the learned reward model guides policy optimization. Our main theorem shows that this two‑phase design and the MLE optimization of the reward creates the misalignment between the update gradient and policy gradient maximizing the true human values. Although this issue affects both PPO and DPO equally, we present our theorem in the context of DPO for clarity. We articulate this motivation in the Introduction and provide a detailed discussion of its implications for PPO in Appendix G. By the same reasoning, our theoretical principle extends to IPO; however, because IPO modifies the optimization objective (and thus the gradient), its optimal sampling scheme differs from that derived for DPO. For clarity and focus, we leave a detailed analysis of IPO and its corresponding optimal sampling scheme to future work.
>
> The contribution of our work is multifaceted. First, we rigorously identify and characterize the misalignment problem inherent to the two‑phase RLHF framework—an issue largely overlooked by prior work. Second, we develop a comprehensive, assumption‑light theoretical analysis that directly yields an optimal sampling strategy; unlike Cen et al. (VPO), our approach makes no restrictive assumptions (e.g., reward‑model linearity) and therefore holds under very general conditions. This requires substantial theoretical innovation. Third, we empirically validate PILAF's effectiveness on modern large language models, demonstrating significant and consistent improvements over existing baselines. Consequently, our contribution extends far beyond proposing a new algorithm: it offers a universal, theory‑driven perspective on addressing misalignment for RLHF.
>
> Finally, we respectfully disagree that DPO is not state‑of‑the‑art. Recent studies confirm that the principal driver of performance in RLHF is whether an algorithm incorporates online data generation. When implemented online, DPO matches PPO’s performance as reported by [Noukhovitch et al. 2025, Tang et al. 2024]. Accordingly, we evaluate our method in both iterative and online settings.
>
> Noukhovitch, Michael, et al. "Asynchronous RLHF: Faster and More Efficient Off-Policy RL for Language Models." ICLR 2025.
>
> Tang, Yunhao, et al. "Understanding the performance gap between online and offline alignment algorithms." arXiv preprint arXiv:2405.08448 (2024).
>
> Please let us know if our responses address your concerns.

---

> > ### Comment · Reviewer_kgC8 · 2025-04-03
> >
> > I appreciate the authors' response.
> > 1. I was suggesting including more experiments about applying the new sampling mechanism in other algorithms like PPO to verify its generality.
> > 2. As for the performance of online DPO and PPO, Tang et al's work didn't really implement PPO and study its performance. Noukhovitch et al's work showed that online DPO is slightly better than a baseline PPO implementation on GSM8K. However, I noticed some recent, more extensive comparisons (https://github.com/RLHFlow/Online-DPO-R1) showing that PPO is still the strongest algorithm on more datasets. I guess there are some controversies going on, so I just suggest that the authors can try this sampling mechanism in PPO too. If it works out, this would make a greater impact.
> > Overall, I decided to maintain my score.

---

### Official Review · Reviewer_mZPC · 2025-03-12

**Overall Recommendation:** 3

**Summary:**

The paper "PILAF: Optimal Human Preference Sampling for Reward Modeling" introduces Policy-Interpolated Learning for Aligned Feedback (PILAF), a novel sampling strategy designed to improve reinforcement learning from human feedback (RLHF), particularly in reward modeling for aligning large language models (LLMs) with human values.

**Claims And Evidence:**

See weaknesses.

**Essential References Not Discussed:**

No.

**Experimental Designs Or Analyses:**

See Methods And Evaluation Criteria.

**Methods And Evaluation Criteria:**

1. The authors simply use HH-RLHF dataset, which is not enough to validate their conclusion. It would be better to execute on more benchmarks.

2. The approach is online iterative DPO, but they simply compare with the vanilla DPO, which is unfair, because online DPO outperforms the vanilla one. The authors are suggested to compare their algorithm with more online methods, such as the general online DPO [1,2,3]

Besides, the works [1,2] are actually online DPO instead of simply iterative DPO since they collect samples generated from the trained policy and get them labeled by a preference oracle.


[1] Xiong, W., Dong, H., Ye, C., Wang, Z., Zhong, H., Ji, H., Jiang, N., and Zhang, T. Iterative preference learning from human feedback: Bridging theory and practice for rlhf under kl-constraint. In Forty-first International Con- ference on Machine Learning, 2024.
[2] Dong, H., Xiong, W., Pang, B., Wang, H., Zhao, H., Zhou, Y., Jiang, N., Sahoo, D., Xiong, C., and Zhang, T. Rlhf workflow: From reward modeling to online rlhf, 2024.
[3] Guo, S., Zhang, B., Liu, T., Liu, T., Khalman, M., Llinares, F., Rame, A., Mesnard, T., Zhao, Y., Piot, B., et al. Direct language model alignment from online ai feedback. arXiv preprint arXiv:2402.04792, 2024.

**Other Comments Or Suggestions:**

See comments above.

**Other Strengths And Weaknesses:**

No.

**Questions For Authors:**

Please answer the questions above.

**Relation To Broader Scientific Literature:**

They provide realistic ways for online exploration for RLHF.

**Theoretical Claims:**

The theoretical analysis is a little confusing and it is hard to get the intuitions why they use such sampling strategy \pi- and \pi+. The authors are suggested to clarify the insights more clearly instead of just listing theorems and equations that seems distinct from the algorithms. Now, it's hard to find connections between the theorem and experiments.

---

> ### Author Rebuttal · Authors · 2025-04-01
>
> We thank the reviewer for their time in providing the review.
>
> 1. > HH-RLHF not enough. More benchmarks.
>
> Please allow us to put our work into more context. Our contributions extend beyond simply empirically validating a new sampling algorithm. Rather, first, we identify and rigorously characterize a previously overlooked misalignment problem in RLHF. Second, we develop a general theoretical framework—free of restrictive assumptions—that directly informs a principled solution for addressing the problem with guarantees. Third, we translate these insights into T‑PILAF, a practical algorithm which we validate at scale using 8B LLMs and the HH‑RLHF dataset. Together, we believe the novel theoretical results and large‑scale experiments convincingly demonstrate (1) the existence of misalignment in standard RLHF, (2) how it can be addressed in a first‑principles manner, and (3) the effectiveness of our method in practice. For further discussion of our contributions, please refer to response #3 to Reviewer kgC8.
>
> We believe the strength of our proposed algorithm lies in its theoretical grounding, provably solving the alignment problem that we expose. As such, we feel that validation on a prominent benchmark, HH-RLHF, with a large LLM summoning a significant amount of compute at our disposal, is convincing evidence.
>
> 2. > The approach is online iterative DPO, but they simply compare with the vanilla DPO, which is unfair, because online DPO outperforms the vanilla one. The authors are suggested to compare their algorithm with more online methods, such as the general online DPO [1,2,3]
>
> Following the definitions in [1,3], we distinguish between iterative and online data collection as follows: iterative sampling generates all preference data from the policy network at the beginning of each iteration, whereas online sampling produces preference data continuously, with each new batch drawn from the current policy network.
>
> Importantly, we are not comparing vanilla DPO against these modes; instead, we hold the underlying RLHF iterative/online framework fixed and vary only the sampling strategy. Concretely, in both the iterative and online setups, every method collects preference data from the policy network—iteratively in the iterative setup (Sections 6.1) and batch‑by‑batch in the online setup (Section 6.2)—with the sole difference being how those samples are generated. This distinction is detailed in the implementation paragraphs and via different values of $n_t$ in Algorithm 1. The term *Vanilla* only denotes the sampling method as vanilla, as shown in Table 1. Thus, we are ensuring a fair comparison exactly as the reviewer suggested.
>
> 3. > Besides, the works [1,2] are actually online DPO instead of simply iterative DPO since they collect samples generated from the trained policy and get them labeled by a preference oracle.
>
> We adopt the definitions of “online” and “iterative” sampling exactly as presented in the referenced papers. In particular, collecting samples from a fully trained policy at each iteration is referred to as iterative DPO in [1, 2].
>
> 4. > The theoretical analysis is a little confusing and it is hard to get the intuitions why they use such sampling strategy \pi- and \pi+. The authors are suggested to clarify the insights more clearly instead of just listing theorems and equations that seems distinct from the algorithms. Now, it's hard to find connections between the theorem and experiments.
>
> To provide an intuition for the misalignment problem, note that DPO implicitly defines the reward as $r_\theta(x,\vec y) = \beta \cdot \log\left(\frac{\pi_\theta(\vec y \mid x)}{\pi_{\mathrm{ref}}(\vec y \mid x)}\right)$ which is trained via maximum‑likelihood estimation. When training with preference data generated by $\pi_\theta$, the optimization is biased. Reviewer PDrJ also summarizes this aptly as "it may not generalize well to reflect true preferences because the sampled comparisons do not represent the broader preference landscape." Nonetheless, this intuition does not prescribe a concrete sampling strategy for correcting the misalignment.
>
> Instead, we use theoretical analysis to reveal the root cause and derive a principled solution. By comparing Equations 14 and 15a, we demonstrate how the gradient produced by standard sampling diverges from the true alignment gradient. Following Reviewer qMsM’s suggestion, we have added a lemma in the main text that explicitly shows this discrepancy. These insights directly inform T‑PILAF, a sampling algorithm that leverages $\pi^-$ and $\pi^+$, to realign the empirical gradient with its theoretical counterpart (as explained in line 245 left). PILAF is then the practical instantiation of T‑PILAF, and our experiments validate its effectiveness. This seamless integration of theory and practice is precisely what Reviewer PDrJ praised: “these theoretical results motivate the practical design of PILAF.”
>
> Please let us know if our responses address your concerns.

---

### Official Review · Reviewer_PDrJ · 2025-03-17

**Overall Recommendation:** 4

**Summary:**

This paper investigates strategies to leverage interpolated response sampling for improving human preference data collection and reward modeling in RLHF. The authors propose a Policy-Interpolated Learning for Aligned Feedback PILAF method that generates response pairs by interpolating between a reference policy and the current policy to better align reward model training with the true preference objective; then, they develop a practical version of PILAF and evaluate it in iterative and online DPO training setups. They find significant gains in reward model performance, alignment quality, and sample efficiency compared to Vanilla sampling and Best-of-N sampling methods.

**Claims And Evidence:**

I think the mathematical proof and experiments together provide a reasonable justification for the effectiveness of the proposed method. Although more analysis could strengthen the connection, the current results are generally convincing.

**Essential References Not Discussed:**

Not found.

**Experimental Designs Or Analyses:**

The experiments are extensive, with detailed analysis of the results. These experiments validate the effectiveness of PILAF in improving reward model quality, sample efficiency, and alignment performance and demonstrate the robustness and scalability of the method across different training setups, iterative and online DPO, and model sizes. The evaluation includes comparisons to strong baselines including Vanilla, Best-of-N, and Hybrid sampling, and covers both quantitative metrics, reward model performance, KL divergence, and training dynamics.

**Methods And Evaluation Criteria:**

The idea of interpolated response sampling for improving reward modeling in RLHF makes sense for the problem. Most existing work focuses on sampling responses directly from the current policy or using simple heuristics, but these approaches cannot effectively align reward model learning with the true human preference objective. This could limit the efficiency and quality of preference data. This paper introduces a method that interpolates between a reference policy and the current policy to generate more informative and aligned comparisons, as well as significantly improves reward model quality and sample efficiency.

The workflow is well-structured, as it builds on theoretical insights T-PILAF and adapts them into a practical algorithm PILAF to make the method applicable to real-world RLHF pipelines. This approach balances exploration and exploitation during response sampling and further enhances the alignment between reward model training and human preferences.

**Other Comments Or Suggestions:**

The authors may consider adding an ablation study to analyze whether the gains from PILAF are truly due to its interpolated sampling mechanism and gradient alignment rather than other confounding factors. For example, comparing PILAF with versions that remove or vary the interpolation component would help validate this core contribution.

**Other Strengths And Weaknesses:**

Thank you for taking the time to write and submit this work. The key strengths that I have observed are as follows:
1. I really appreciated the effort that the authors put into the related works section. They clearly did their research into the relevant domains and introduced me to new papers as well.
2. Their preliminary and motivation works section was very clear. I especially appreciated how they took the time to lay out the theoretical formulation and explain the connection between reward modeling gradients and preference sampling in detail.
3. Figure 1 is very clear. it illustrates the core idea of this method.

Overall, the authors bring up an interesting problem of aligning reward model training with true human preference gradients. In reward modeling, this would be an interesting setting to see how we can optimize preference data collection strategies to improve alignment given the constraints of human annotation cost and model sample efficiency.

**Questions For Authors:**

See above.

**Relation To Broader Scientific Literature:**

This paper is related to preference-based RL, reward modeling in RLHF, and sampling-based data efficiency methods. The main differences are that (1) it introduces interpolated sampling to address gradient misalignment (against standard preference data collection), (2) it provides a formal analysis of gradient alignment (against heuristic sampling approaches), and (3) it focuses on optimizing sample efficiency through theory-grounded methods rather than relying solely on empirical strategies.

**Theoretical Claims:**

The authors provide two main parts of theoretical claims.

 (1) The authors formalize the oracle objective that an ideal reward model should be optimized to reflect true human preferences. They analyze how standard response sampling strategies, e.g., sampling only from the current policy, cause a gradient misalignment between the reward model's learning objective and the oracle objective. This misalignment means that even if a reward model fits the data it sees, it may not generalize well to reflect true preferences because the sampled comparisons do not represent the broader preference landscape.

(2) The authors propose T-PILAF. This framework generates response pairs by interpolating between a reference policy and the current policy. They prove that this interpolation mechanism aligns the reward model's gradient with the oracle gradient, thus correcting the bias from single-policy sampling. The proof shows how interpolated sampling balances exploration and exploitation. In this way, the reward model training is more statistically efficient and better aligned with the underlying preference function. Overall these theoretical results motivate the practical design of PILAF.

---

> ### Author Rebuttal · Authors · 2025-04-01
>
> Thanks for appreciating our work, especially its development from theory to algorithm design. We are sincerely pleased that the reviewer aacknowledged the misalignment problem we identified, found the combination of mathematical proofs and experiments to provide a reasonable justification, and considered our experimental results to be extensive.
>
> 1. > The authors may consider adding an ablation study to analyze whether the gains from PILAF are truly due to its interpolated sampling mechanism and gradient alignment rather than other confounding factors. For example, comparing PILAF with versions that remove or vary the interpolation component would help validate this core contribution.
>
> Thank you for the suggestion. Following it, we added two ablation studies to isolate the contributions of PILAF’s interpolation and extrapolation components. Each component was replaced individually with vanilla sampling, yielding two baselines: one with $({\vec y}^a, {\vec y}^b)=(\pi_\theta^+, \pi_\theta)$ (ablation of the interpolation component) and one with $({\vec y}^a, {\vec y}^b)=(\pi_\theta, \pi_\theta^-)$ (ablation of the extrapolation component). We denote these ablation variants as PILAF-extrapolate and PILAF-interpolate, where one response is obtained via vanilla sampling and the other via extrapolation or interpolation, respectively. Due to time constraints, we completed these ablations only for the online setup; we plan to extend this to the iterative setting in the camera-ready version.
>
> We include the figures at [anonymous URL](https://drive.google.com/file/d/11wfcQT07dxy0IpXuIcdRUrtRBoPPuwXD). Our theory suggests that the two sampling responses should come from different distributions in order to yield a controlled difference that the model can effectively learn from. Both ablation variants introduce such differences and outperform vanilla sampling. However, the variant with only interpolation (combined with vanilla sampling for the other response) performs much worse than full PILAF, highlighting the importance of the extrapolation response. The PILAF-extrapolate variant achieves slightly worse final results, and its convergence is much slower (each dot in our figure represents one evaluation after 50 steps). Overall, these ablation results confirm our theoretical prediction that the full PILAF algorithm is the best performing approach.
>
> We thank the reviewer again for their time and constructive feedback.

---

> > ### Comment · Reviewer_PDrJ · 2025-04-04
> >
> > I appreciate the authors’ response. I will keep the score and I do like this work.

---

### Official Review · Reviewer_qMsM · 2025-03-22

**Overall Recommendation:** 2

**Summary:**

This paper introduces a sampling strategy for collecting human preference data in RLHF (specifically, DPO) setting. It aims to align preference-based reward modeling with the true (oracle) objective by interpolating between the current and reference policies during response generation. Theoretical analysis shows that the proposed aligns gradients of the true (oracle) objective in the first order, making training more consistent and efficient. The authors validate this in both iterative and online DPO settings, demonstrating the effectives regard the DPO loss.

**Claims And Evidence:**

In general, the motivation of this paper is clear and the claims and analysis make sense.

**Essential References Not Discussed:**

N/A

**Experimental Designs Or Analyses:**

Yes, although the experiments on DPO is not comprehensive compared to other papers, it supports the claim that the algorithm better aligns with the DPO loss regarding the true objective.

**Methods And Evaluation Criteria:**

Yes.

**Other Comments Or Suggestions:**

N/A

**Other Strengths And Weaknesses:**

Strength is that this problem study a important aspect of RLHF, that is how to better sample data in the online or iterative DPO setting.

One of the weaknesses is that, it is mentioned that "We show that uniform sampling from the current policy, as is common, leads to misaligned gradients of the two objectives". However, in the theoretical development, I only saw theorem 4.1 showing the result for alignment gradients, but there is no formal results on the claim that uniform sampling results in misaligned gradients. It is not hard to show a theoretical results like this.

**Questions For Authors:**

I have the following questions, and am willing to increase my score if properly addressed:

1. I have a question regarding section 2.3. It is mentioned that the true goal for DPO should be (6). However, the common DPO setting is not very different from (6). It basically replaces $r*$ in (6) by the learned $r_{\theta}$. In other words, the common DPO setting approximates the goal (6) with some errors. I think there should be some results or statement on the analysis between the previous objective and the current objective (6).

2. It was not described how to turn $y^a$ and $y^b$ into $y^w$ and $y^l$. This is problematic when looking at eqn (3).

3. how is the optimal policy $\pi^*$ defined in equation 9?

4. in line 294 and 295, I think it is bettwe to provide a formal statement with proofs or derivation in the appendix.

**Relation To Broader Scientific Literature:**

Related to general alignment of LLMs.

**Theoretical Claims:**

Yes I have checked Theorem 4.1 and 4.2, which look fine to me.

---

> ### Author Rebuttal · Authors · 2025-04-01
>
> We appreciate the reviewer’s insightful comments, which have helped improve the presentation of the misalignment problem. We are also glad that the reviewer enjoyed our theoretical analysis and empirical validation.
>
> 1. > uniform sampling - misaligned gradients
>
> We thank the reviewer for raising this point — a clear, formal statement helps improve our presentation. The discrepancy arises from the difference in gradient formulations between Eqns 14 and 15a.
> To make notations concise, we introduce the following shorthands: $\Delta r^* := r^*(x, y^a) - r^*(x, y^b)$, $\Delta r_{\theta} := r_{\theta}(x, y^a) - r_{\theta}(x, y^b)$ and $g := \nabla_{\theta} r_{\theta}(x, y^a) - \nabla_{\theta} r_{\theta}(x, y^b)$.
>
> **Lemma C.2.**
> $\nabla_{\theta} J(\pi_{\theta}) = \frac{1}{2 \beta} E_{y^a, y^b \sim \pi_{\theta}(\cdot | x)} [ ( \Delta r^* - \Delta r_{\theta} ) g]$.
>
> **(Corollary of) Lemma C.3.** For vanilla response sampling scheme, $\nabla_{\theta} L(\theta) = -E_{y^a, y^b \sim \pi_{\theta}(\cdot | x)} [(\sigma(\Delta r^*)-\sigma(\Delta r_{\theta}))g]$.
>
> These two gradients share a similar structure. The key difference is $\Delta r^* - \Delta r_{\theta}$ for $\nabla_{\theta} J(\pi_\theta)$ and $\sigma(\Delta r^*) - \sigma(\Delta r_{\theta})$ for $\nabla_{\theta} L(\theta)$.
> To correct for this mismatch, T-PILAF adjusts the response sampling distribution: It reweights the pairwise response sampling so that the density ratio between the vanilla scheme and T-PILAF approximates the derivative $\sigma'(\Delta r_\theta)$. This bridges the gap between the non-linear sigmoid differences and the linear reward differences, leading to better gradient alignment during training.
>
> 2. > Q1: Section 2.3
>
> Let us clarify the root cause of the misalignment. The true objective for both PPO and DPO is to optimize the expected return under the true reward function $r^*$ in (6). In PPO, a reward model $r_\theta$ is trained from human preferences and then used to guide policy optimization. In DPO, $r_\theta$ is learned via $r_\theta(x,y) = \beta \log\frac{\pi_\theta(y | x)}{\pi_{\mathrm{ref}}(y | x)}$. In both frameworks, the final policy’s performance depends on how well $r_\theta$ approximates $r^*$, as you noted.
>
> Prior work assumes that on‑policy samples from $\pi_\theta$ suffice for improving both the policy and its reward model. We show this is incorrect: gradients from standard on‑policy sampling do not align with the true policy gradient under $r^*$, limiting  $r_\theta$’s ability to close the approximation gap. As Reviewer PDrJ perfectly summarizes: "it may not generalize well to reflect true preferences because the sampled comparisons do not represent the broader preference landscape." This applies to both PPO and DPO; we present the result in the DPO setting for clarity.
>
> In contrast, our T‑PILAF sampling scheme is designed to align the empirical gradient with the true policy gradient under $r^*$. This ensures each update of $r_\theta$ approximates $r^*$ in the optimal first-order direction. Our statistical results also shows T-PILAF minimizes variance. All of our results focus on comparing between the empirical objective and the true objective in (6), showing precisely how vanilla sampling produces misaligned gradients when updating the reward model.
>
> 3. > How to turn $y^a$ and $y^b$ into $y^w$ and $y^l$.
>
> Before Eqn (1), we noted that $y^w$ and $y^l$ were human-annotated as the preferred and unpreferred responses, respectively. Eqn. (1) introduced the commonly used Bradley-Terry (BT) model, and we stated explicitly that the BT assumption was adopted throughout this paper.
>
> 4. > Definition of $\pi^*$ in eqn 9
>
> The ground-truth reward function in the BT model (Eqn 1) is denoted by $r^*$. Then, the notation $\pi^*$ refers to the optimal policy that maximizes the value function $J(\pi)$, as in Eqn (6):
> \begin{align*}
>  \pi^*(y | x) = \frac{1}{Z(x)} \pi_{ref}(y | x) \exp(\frac{1}{\beta} r^*(x,y)).
> \end{align*}where $Z(x)$ is a partition function that ensures $\pi^*(\cdot | x)$ sums to 1.
>
> 5. > line 294 and 295, proofs.
>
> Following the suggestion, we add the derivation:
>
> **Proof:**
>
> Starting from
> $\pi_\theta^+(y_t | x, y_{1:t-1}) = \frac{1}{Z(x, y_{1:t-1})} \pi_\theta(y_t | x, y_{1:t-1}) \big(\frac{\pi_\theta(y_t | x, y_{1:t-1})}{\pi_{ref}(y_t | x, y_{1:t-1})}\big)^\beta$,
>
> we rewrite it as:
> $\pi_\theta^+(y_t | x, y_{1:t-1}) \propto \exp\big((1+\beta)\log \pi_\theta(y_t | x, y_{1:t-1}) - \beta \log \pi_{ref}(y_t | x, y_{1:t-1})\big)$.
>
> Define the logits:
> $h_\theta(y_t | x, y_{1:t-1}) = \log \pi_\theta(y_t | x, y_{1:t-1})$,
> $h_{ref}(y_t | x, y_{1:t-1}) = \log \pi_{ref}(y_t | x, y_{1:t-1})$.
>
> Then
> $\pi_\theta^+(y_t | x, y_{1:t-1}) \propto \exp((1+\beta) h_\theta(y_t | x, y_{1:t-1}) - \beta h_{ref}(y_t | x, y_{1:t-1})).$
>
> Normalizing over all $y_t$ leads to the softmax form:
> $\pi_{\theta,\beta}(\cdot | x, y_{1:t-1}) = \mathrm{softmax}((1+\beta) h_\theta - \beta h_{ref})$.
>
> Please let us know if our responses address your concerns.

---

### Decision · Program_Chairs · 2025-05-01

**Decision:**

Accept (poster)

**Comment:**

This paper introduces PILAF (Policy-Interpolated Learning for Aligned Feedback), a sampling strategy for collecting human preference data in DPO. The idea is to interpolate between the current and reference policies during response generation to better align reward model training with the true preference objective. The reviewers recognized the novel sampling strategy, the contribution of theoretical analysis, and the positive experimental results in iterative and online DPO settings. The reviewers also raised concerns and suggestions about additional experiments and comparisons to literature and baselines, which are mostly addressed in rebuttal. All reviewers recommend accepting the paper during the discussion phase, and the AC agrees with the recommendation.